# Heavy-atom tunnelling in singlet oxygen deactivation predicted by instanton theory with branch-point singularities

Imaad M. Ansari [1], Eric R. Heller [1,2], George Trenins [1,3] & Jeremy O. Richardson [1] ✉

The reactive singlet state of oxygen ($O_2$) can decay to the triplet ground state nonradiatively in the presence of a solvent. There is a controversy about whether tunnelling is involved in this nonadiabatic spin-crossover process. Semiclassical instanton theory provides a reliable and practical computational method for elucidating the reaction mechanism and can account for nuclear quantum effects such as zero-point energy and multidimensional tunnelling. However, the previously developed instanton theory is not directly applicable to this system because of a branch-point singularity which appears in the flux correlation function. Here we derive a new instanton theory for cases dominated by the singularity, leading to a new picture of tunnelling in nonadiabatic processes. Together with multireference electronic-structure theory, this provides a rigorous framework based on first principles that we apply to calculate the decay rate of singlet oxygen in water. The results indicate a new reaction mechanism that is 27 orders of magnitude faster at room temperature than the classical process through the minimum-energy crossing point. We find significant heavy-atom tunnelling contributions as well as a large temperature-dependent $H_2O/D_2O$ kinetic isotope effect of approximately 20, in excellent agreement with experiment.

The first excited electronic state of $O_2$, a singlet with $^1\Delta_g$ symmetry, is a reactive species that plays an important role in atmospheric and environmental chemistry[1–3] and has wide-ranging applications[4–6] from synthetic chemistry[7,8] to photodynamic cancer therapy[9]. Its reactivity is greatly influenced by the timescale of its decay to the less reactive triplet $^3\Sigma_g^-$ state, which can take place via radiative and nonradiative processes[10,11]. In the gas phase, both processes are formally forbidden by symmetry: the radiative transition is electric-dipole and spin forbidden[12], while for the nonradiative transition, the spin–orbit coupling is zero[13]. This is reflected in the long reported lifetimes on the order of minutes[10,14,15], principally determined by the radiative process.

In contrast, in the condensed phase, the nonradiative channel dominates, with lifetimes on the order of μs in $H_2O$, ms in $C_2Cl_4$ to a few tenths of a second in $CCl_4$[10,16,17]. Phenomenologically, this can be understood in terms of collisions and the resulting energy transfer between $O_2$ and solvent molecules. This is the physical justification behind the electronic-to-vibrational (e-to-v) energy-transfer model[10,18,19], where $O_2$ is said to decay to its triplet state by transferring its electronic energy to the vibrational modes of the solvent molecule. However, this simplistic description of the solvent solely as an energy sink ignores the effect the solvent can have on the electronic structure of $O_2$[13,17,20]. It is this interaction that facilitates a spin crossover (also known as intersystem crossing) by breaking the symmetry

[1]Department of Chemistry and Applied Biosciences, ETH Zürich, 8093 Zürich, Switzerland. [2]Present address: Department of Chemistry, University of California, Berkeley, 94720 Berkeley, USA. [3]Present address: MPI for the Structure and Dynamics of Matter, Luruper Chaussee 149, 22761 Hamburg, Germany. ✉e-mail: jeremy.richardson@phys.chem.ethz.ch

thereby introducing a non-zero spin–orbit coupling between the singlet and triplet states.

Given the typically small values of the spin–orbit coupling[21], it is often valid to use Fermi's Golden Rule (FGR)[22,23] to calculate the rate constant for this nonadiabatic process. Although FGR simplifies the full problem by separating the Hamiltonian into independent reactant and product parts, it still requires solving the vibrational Schrödinger equation for these sub-problems, which is computationally infeasible for multidimensional molecular systems.

Classical approximations to FGR include Marcus theory[24] and nonadiabatic transition-state theory (NA-TST)[25], but such theories will not allow us to test the contribution of tunnelling. It has been proposed that tunnelling plays a significant role in the solvent-mediated nonradiative decay, as evidenced by a large H/D kinetic isotope effect (KIE) of about 20 in a range of solvents including water, methanol, benzene, and toluene[11,16]. Furthermore, this effect has been observed to increase at lower temperatures, leading Jensen, Ogilby and coworkers[11] to suggest that tunnelling might play a role in the decay.

However, in a later theoretical study from the same group, the authors could not find evidence of significant tunnelling contributions[20]. This study considered a 1:1 complex between $O_2$ and a solvent molecule as the reactive species and was based on the assumption that the reaction proceeds through its singlet–triplet minimum-energy crossing point (MECP), which is the analogue of a transition state for nonadiabatic reactions. They acknowledged that their tunnelling correction was based on a one-dimensional reaction coordinate connecting the singlet minimum and the MECP; it does not therefore account for corner-cutting effects, i.e., tunnelling mechanisms that do not pass through the MECP. As it is known that corner cutting can significantly alter the reaction mechanism and the resulting rate, they suggested that a more accurate description of the tunnelling would therefore require a more involved multidimensional treatment.

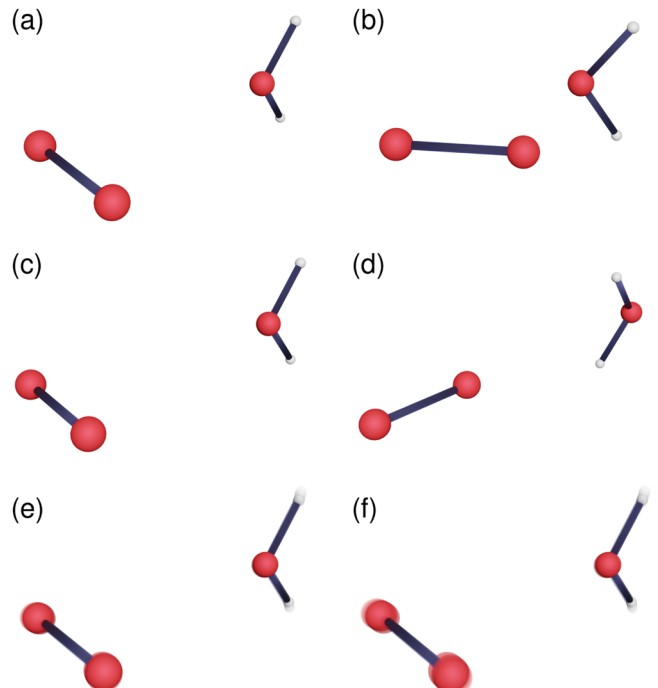

**Fig. 1 | Key geometries of the oxygen–solvent complex. a** The reactant minimum ($C_{2v}$), **b** the minimum-energy crossing point ($C_s$), **c** a product saddle point ($C_{2v}$), **d** the product minimum ($C_{2v}$), the branch-point instanton of (**e**) $O_2{\cdots}H_2O$ ($C_{2v}$) and (**f**) $O_2{\cdots}D_2O$ at 300 K ($C_{2v}$). Atomic tunnelling in (**e**) and (**f**) is depicted by a motion blur. Key geometrical information (i.e., bond lengths and angles) for the structures is presented in Fig. 3. The geometries are provided as a Source Data file.

Golden-rule instanton theory[26] is a method for calculating FGR rates from first principles that takes into account quantum nuclear effects such as multidimensional tunnelling and zero-point energy. It is rigorously derived from the path-integral formulation[27] of quantum mechanics using a semiclassical approximation based on a single periodic classical trajectory in imaginary time, called an instanton. The instanton describes the dominant tunnelling pathway in multi-dimensional space and includes a hop from one potential energy surface (PES) to the other, providing an elegant mechanistic description of the nonadiabatic tunnelling process that takes into account corner cutting.

The decay of singlet oxygen takes place in the Marcus inverted regime[24], where the minima are on the same side of the MECP. Golden-rule instanton theory has been recently generalised to this regime[28]. The key difference with the instanton in the normal regime is that in the inverted regime, part of the instanton travels in negative imaginary time. We have successfully applied golden-rule instanton theory in first-principles simulations of two nitrenes[29] in the normal regime and to thiophosgene[30] in the inverted regime, where along with excellent agreement with experimental rates, analysis of the tunnelling pathway provided a physical explanation of the observed KIEs.

In our previous work, the instanton was defined as the path which makes the action stationary, and locating it involves a saddle-point search in the space of paths[31]. In the inverted regime, this corresponds to a high-index saddle point[28]. However, our initial attempts at applying golden-rule instanton theory to the decay of singlet oxygen resulted in a trajectory that corresponded to a saddle point of the wrong index. The reason for this discrepancy was traced to the appearance of a singularity of the flux correlation function inside the integration contour. Cauchy's integral theorem[32] can no longer be used and the standard golden-rule instanton rate expression is no longer valid.

A methodological extension is therefore necessary to treat this problem. In the Methods section, we present alternative integration contours and the corresponding asymptotic approximations to the rate, followed by a preliminary benchmark on a model system. We then combine these methods with ab initio electronic-structure theory in the 'Results' section to calculate the nonradiative decay rate of singlet oxygen interacting with a water molecule. We discover evidence of heavy-atom tunnelling and explain the experimentally observed KIEs.

## Results

Following previous theoretical work[20], we studied the complex formed by oxygen interacting with a single water molecule, denoted $O_2{\cdots}H_2O$. The isotopologues $O_2{\cdots}D_2O$ and $^{18}O_2{\cdots}H_2O$ were also studied. Calculations were performed over a range of temperatures between 275 K and 330 K in conjunction with on-the-fly ab initio multireference electronic-structure theory (with full details presented in the Methods section).

### Single-point geometries

The optimised structures corresponding to the minima of the singlet (reactant) and triplet (product) states as well as the MECP are shown in Fig. 1a–d. While both minima belong to the $C_{2v}$ point group, the O–O bond is orthogonal to the principal axis in the singlet but parallel to it in the triplet. The relative orientation of $H_2O$ is also flipped. In addition, a product saddle point was found, which has a structure similar to the reactant minimum with $C_{2v}$ symmetry.

In contrast, the MECP only has one symmetry plane and belongs to the $C_s$ point group. Unlike the minima, where $O_2$ and $H_2O$ interact primarily via long-range interactions, at the MECP structure they are in close contact, such that there is significant orbital interaction between them. In fact, at the MECP geometry, the orbitals localised on $H_2O$ contribute significantly to the natural orbitals that correspond to the active space.

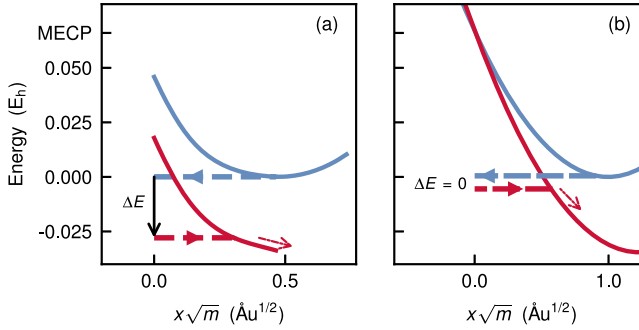

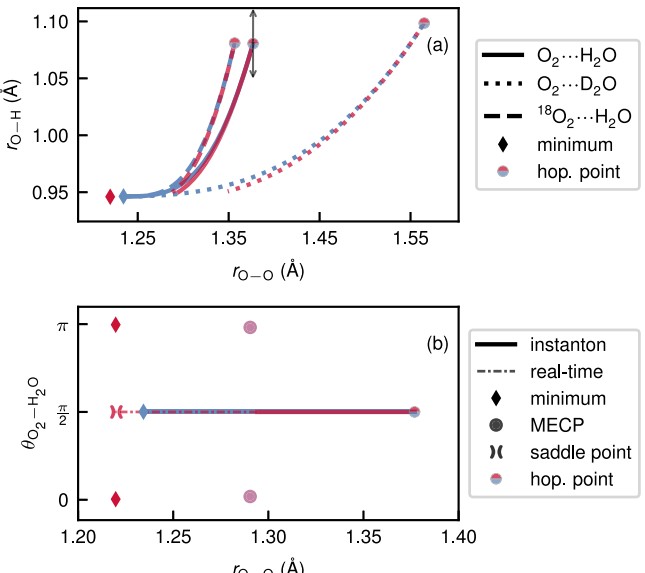

**Fig. 2 | An illustration of the instanton tunnelling pathways on the potential energy surfaces (PESs). a** The branch-point instanton for $O_2 \cdots H_2O$ at 300 K and (**b**) a stationary-action instanton for the model system [Eq. (15)], plotted against mass-weighted path length. The instanton is represented by dashed lines and the PESs with solid lines. Blue corresponds to the reactant and red to the product. The PESs to the right of the endpoints of the instanton segments in (**a**) were constructed by extrapolation, and the tunnelling energies in (**b**) are spaced slightly apart for illustration purposes only. The vertical arrow in (**a**) indicates the energy difference between the two trajectories while the slanted red arrow denotes relaxation towards the product minimum after tunnelling. Data for (**a**) are provided as a Source Data file.

**Fig. 3 | A projection of branch-point instantons for $O_2 \cdots H_2O$ and its iso-topologues at 300 K. a** Along the dominant tunnelling modes, i.e. the $O_2$ bond length, $r_{O-O}$, and the O–H (or O–D) bond distance in the solvent molecule, $r_{O-H}$ and (**b**) along $r_{O-O}$ and the symmetry-breaking angle between the O–O and the sum of the two O–H bond vectors, $\theta_{O_2-H_2O}$. The real-time minimum-energy decay path of $O_2 \cdots H_2O$ after tunnelling is shown as a narrow dash-dotted line. The hopping points and minima are marked in (**a**) while in (**b**) the MECP and the saddle point are also marked. The projection of the branch-point eigenvector for $O_2 \cdots H_2O$ at the hopping points is shown in (**a**) with black arrows. In both plots, the colour denotes the reactant state (blue) or product state (red). Note that all data shown in (**b**) is mirrored in the range $[-\pi, 0]$. The geometries and mass-weighted branch-point eigenvector are provided as Source Data files.

A classical theory, such as NA-TST, would assume that the reaction proceeds through the MECP. However, in the following, we will demonstrate that this is not the case, due to strong tunnelling effects.

## Instanton tunnelling mechanism

Rigorous chemical reaction rate theory[33] defines the rate constant in terms of the time integral over the flux correlation function, $c_{ff}$. In instanton theory[34], we make a semiclassical approximation to $c_{ff}$ at imaginary time, $\tau$, from which we obtain a prediction for the rate constant. The semiclassical approximation involves locating the tunnelling path of least action and treating fluctuations around this path to second order. This results in the formula $c_{ff} \simeq \sqrt{1/\Theta(\tau)}\, e^{-\phi(\tau)/\hbar}$, where $\phi$ is the action along the tunnelling pathway and $\Theta$ encodes the fluctuations.

Standard instanton theory[34] is based on the principle of stationary action, i.e., $d\phi/d\tau = 0$. However, in this case, the Hessian of the stationary-action instanton was found to have the wrong number of negative eigenvalues, which implies the existence of a singularity of the flux correlation function in the complex plane. One possible solution is to instead make the approximation around the point $dc_{ff}/d\tau = 0$ (see Methods). Although this stationary-flux instanton theory gives similar predictions to the method we now describe, it does not enjoy the same mathematical rigour.

We show that the singularity may be caused by significant differences in vibrational frequencies in the reactant and product states, which allows an infinite number of tunnelling paths to contribute equally to the reaction rate. As explained in the Methods section, we have developed a new approach called branch-point instanton theory which accounts for the singularity in the rate calculation. Here the mechanism is dominated by the singularity and the relevant path is the one for which $\Theta(\tau) = 0$. This approach was then applied using the ring-polymer formulation to simulate the nonradiative decay of singlet oxygen in water.

In Fig. 2, we depict the two types of instantons. Unlike the stationary-action instanton, the branch-point instanton does not conserve classical energy across the hop, i.e., the energies of the reactant and product trajectories are not equal. This allows it to hop at a point below the crossing seam, which reduces the action and increases the tunnelling rate.

The branch-point instanton path provides insight into the reaction mechanism. Our simulations predict substantial tunnelling along

the $O_2/^{18}O_2$ and $H_2O/D_2O$ symmetric stretching modes [Fig. 1e, f and Fig. 3a]. It is particularly noteworthy that even the heavy oxygen atoms are involved in tunnelling at room temperature. Surprisingly, the tunnelling pathway was the longest for $O_2 \cdots D_2O$ [Fig. 3a]. This goes against the conventional understanding of tunnelling, where heavier particles are expected to have shorter tunnelling paths. This unexpected behaviour is a consequence of our new tunnelling mechanism, which is dominated by the branch-point rather than the stationary-action path. In particular, at 300 K, the branch-point singularity is located at $\tau_{BP}/\beta\hbar = -0.61$ for $O_2 \cdots D_2O$ but at $-0.44$ for $O_2 \cdots H_2O$. The amount of imaginary time spent on the product state is $\tau_{BP}$ and on the reactant state is $\beta\hbar - \tau_{BP}$, which implies that $O_2 \cdots D_2O$ has an overall longer time to tunnel on both the reactant and product state (considering the absolute magnitudes) and can therefore travel a longer distance. Normally, atoms will only follow a longer tunnelling path if it is associated with a lower action and a higher tunnelling probability[35]. However, these rules do not apply in this case and in fact the action of $O_2 \cdots D_2O$ ($\phi_{BP}/\hbar = 19.18$) is larger than that of $O_2 \cdots H_2O$ ($\phi_{BP}/\hbar = 15.01$), such that the lighter atoms are more likely to tunnel as expected from the quantum–classical correspondence principle.

In contrast, substituting $^{16}O_2$ with $^{18}O_2$ results in the standard behaviour, i.e., the tunnelling path along the $O_2$ stretch is shortened slightly. The value $\tau_{BP}/\beta\hbar$ only differs marginally from that of $O_2 \cdots H_2O$ and results in a slightly larger action ($\phi_{BP}/\hbar = 15.12$ for $T = 300$ K). We will shortly see how these results explain the trends seen in the KIEs.

Previous studies on similar $O_2$–solvent complexes found that using a one-dimensional tunnelling correction along the intrinsic reaction coordinate, which passes through the MECP along the $O_2$ stretch mode and the intermolecular stretch, was unable to account for experimental data[20]. However, the onset temperature (below

**Table 1 | Spin-crossover rate constants, $k_2$, in units of ms$^{-1}$, calculated with nonadiabatic transition-state theory (NA-TST), as well as stationary-flux (SF) and branch-point (BP) instanton theory**

| System | $k_{NA\text{-}TST}$ | $k_{SF}$ | $k_{BP}$ |
|---|---|---|---|
| $O_2 \cdots H_2O$ | $3.5 \times 10^{-26}$ | 20.6 | 21.3 |
| $O_2 \cdots D_2O$ | $5.0 \times 10^{-26}$ | 0.87 | 1.02 |
| $^{18}O_2 \cdots H_2O$ | $3.4 \times 10^{-26}$ | 16.8 | 17.7 |

All calculations were performed at 300 K.

which tunnelling effects are expected to be important)[29] for this reaction was found to be about 2300 K, suggesting a large tunnelling contribution at room temperature. This is confirmed by branch-point instanton theory, which predicts a dramatic 27 orders of magnitude speed-up at 300 K when compared to NA-TST in Table 1. Instanton theory locates the optimal tunnelling path in the space of all internal coordinates and is not constrained to pass through the MECP configuration. As shown in Fig. 1, the tunnelling pathway for $O_2 \cdots H_2O$ and $O_2 \cdots D_2O$ differs significantly from the intrinsic reaction coordinate and exhibits significant corner cutting. Passing through the MECP would involve breaking the $C_{2v}$ symmetry of the reactant minimum, whereas the structure of the complex along the tunnelling pathway prefers to maintain its $C_{2v}$ symmetry. The projection of the instanton along the $O_2$ symmetric stretch mode and the angle between the $O_2$ bond vector and the $H_2O$ plane (Fig. 3b) highlights the extent of the corner cutting.

The branch-point eigenvector points in the direction in which the path can deform without changing its action. It is depicted for $O_2 \cdots H_2O$ in Fig. 3a and involves an antisymmetric displacement of the hopping points, which thus corresponds to the antisymmetric family of paths discussed in the Methods section. The vector is dominated by the O–H stretching mode, which implies that the other paths in the family will hop between the singlet and triplet PESs at different O–H bond distances. The frequencies of this mode on the reactant and product surfaces are nearly identical (calculated at the reactant minimum and product saddle points respectively). However, unlike the simple separable model system in the Methods section, where the branch point was a result of differences in frequencies along one degree of freedom, the nuclear degrees of freedom are coupled in the ab initio system, resulting in a curved path and a branch-point eigenvector that changes direction along this path. In such cases, the existence of the branch point cannot be simply explained by frequency differences along a single degree of freedom. Detailed normal-mode analysis and projections of the branch-point eigenvector along the normal modes are provided in Supplementary Fig. 7. This analysis indicates that although the intramolecular modes remain largely unchanged between the two surfaces, the intermolecular modes are significantly more floppy in the product state. It is these frequency differences which are the origin of the branch-point singularity.

Once the system reaches the end of the tunnelling path on the product PES, a downhill descent in the classically allowed region (i.e. in real-time) reveals that the system encounters a saddle point with two imaginary modes (Fig. 3b). The geometry of this saddle point is depicted in Fig. 1c and it maintains the $C_{2v}$ symmetry of the instanton (Fig. 1e) and the reactant minimum (Fig. 1a). The saddle point marks a bifurcation point in the relaxation mechanism, from where the system can relax to either of the two equivalent product minima. Note that the instanton path and the resulting rate constants are completely independent of the downhill descent path and the product saddle point, which are discussed only to understand the symmetry-breaking relaxation of the product.

## Comparison with experiment

Before comparing our results to experiment, it is important to highlight the relationship between the experimentally observed pseudo-first-order rate constants and the rate constant for the decay of the 1:1 complex that we calculate in this work. Such a complex has been used as a model for several theoretical studies of the nonradiative decay of singlet oxygen[10,18,20,36] and can be understood in terms of the following pre-equilibrium condition[11]

$$^1O_2 + H_2O \underset{k_{-1}}{\overset{k_1}{\rightleftharpoons}} {}^1(O_2 \cdots H_2O) \xrightarrow{k_2} {}^3(O_2 \cdots H_2O), \qquad (1)$$

with the steady-state equilibrium constant

$$K_c = \frac{k_1}{k_{-1}} = \frac{[^1(O_2 \cdots H_2O)]}{[^1O_2][H_2O]}, \qquad (2)$$

where square brackets denote concentrations. The overall non-radiative rate constant for the decay of the singlet species, $k_{eff}$, can be obtained from the rate of formation of $^3(O_2 \cdots H_2O)$

$$\frac{d[^3(O_2 \cdots H_2O)]}{dt} = k_2[^1(O_2 \cdots H_2O)] = k_{eff}[^1O_2], \qquad (3)$$

where we have used Eq. (2) to write $k_{eff} = k_2 K_c[H_2O]$. The factor $K_c[H_2O]$ accounts for the probability that $^1O_2$ is found in a complex with $H_2O$. Note that $k_{eff}$ can be related to the nonradiative rate constant reported in previous work[11,20,36] using $k_{nr} = k_{eff}/[H_2O]$.

Our instanton theory approach calculates $k_2$, whereas experiments measure $k_{eff}$. The branch-point instanton results in Table 1 are roughly one order of magnitude smaller than experiment, with reported values of 290 ms$^{-1}$, 14.7 ms$^{-1}$ and 240 ms$^{-1}$ at room temperature for $O_2 \cdots H_2O$[11], $O_2 \cdots D_2O$[11] and $^{18}O_2 \cdots H_2O$[37] respectively. This discrepancy can be accounted for by including the $K_c[H_2O]$ factor from Eq. (2). A rough estimate for $K_c$ can be obtained by calculating the ratio of the partition function of $O_2 \cdots H_2O$ and the product of the partition functions of isolated $O_2$ and $H_2O$, each in their respective minimum-energy geometries and treating them as if it were a gas-phase reaction. Using a value of 55 mol L$^{-1}$ for [H$_2$O], we get $K_c[H_2O] \approx 8$ at 300 K, which predicts a $k_{eff}$ that is of the same order of magnitude as the experimental data. This justifies the instanton approach over NA-TST, which incorrectly predicts a lifetime longer than the age of the universe.

We additionally studied the temperature dependence of the rate constant for rate constant, $k_2$, by applying instanton theory at a range of temperatures between 275 K and 330 K for $O_2 \cdots H_2O$ and between 280 K and 330 K for $O_2 \cdots D_2O$. We only present results below 330 K as it was found that multiple branch points come into play at higher temperatures, which requires an extension of the theory. The results (Supplementary Table 8) show only a very slight change of about 2% for $O_2 \cdots H_2O$ and 27% for $O_2 \cdots D_2O$ across the temperature range. The rate constant for $O_2 \cdots H_2O$ is effectively constant between 275 K and 310 K, indicating that it is in the low-temperature, deep-tunnelling regime.

However, experiments show that increasing the temperature from 278 K to 323 K increases the nonradiative rate by about 9% in H$_2$O and 33% in D$_2$O[11]. We therefore expect that some of the temperature dependence observed in experiments comes from the behaviour of $K_c$. In principle, a molecular dynamics (or path-integral molecular dynamics) simulation of solvated $^1O_2$ would be required to compute the free energy of complexation by explicitly taking into account the disruption of the hydrogen-bonding network when forming the complex. However, this condensed-phase free-energy calculation would be beyond the scope of this work. We instead content ourselves by

demonstrating (in Supplementary Note 2.5) that the enthalpy and entropy of complexation can be fitted to reproduce the experimental data.

In addition to the difficulties of calculating $K_c$ from first principles, the approximation of treating only a single water molecule may not be sufficient for quantitative accuracy in the prediction of the rate constant $k_2$. For instance, in our study, we have neglected many-body effects arising from hydrogen bonding. In principle, we could systematically add more water molecules until the results of the finite cluster converge to those of the true solvated environment. There is some hope that this would not require such a large cluster as there is evidence that the rate-limiting step relies only on the local environment[36].

To provide a direct comparison with the experiment without relying on fitting parameters, we therefore focus on KIEs. Here large parts of the errors associated with the underlying electronic-structure theory and the neglect of many-body effects are expected to cancel out, leaving us with a sensitive probe of the tunnelling contributions to $k_2$. Importantly, we expect $K_c$ to be largely invariant upon isotope substitution, as it is dominated by intermolecular vibrational modes which can be well described by classical statistical mechanics and thus cancel out in the ratio. It is thus appropriate to directly compare KIEs from instanton calculations to experiment, and indeed Table 2 shows excellent agreement, in particular for the large H/D KIE of about 20. In addition, Fig. 4 shows that our predictions match the experimental trends of the temperature-dependent KIE, within the expected numerical error of the instanton calculations.

Additionally, we predict a small $^{18}O_2/O_2$ KIE in water that is greater than unity, consistent with what has been observed experimentally in other solvents[37]. This is expected, as the relative change in mass

**Table 2 | Kinetic isotope effects (KIEs), calculated using non-adiabatic transition-state theory (NA-TST) as well as stationary-flux (SF) and branch-point (BP) instanton methods**

| Systems | NA-TST | SF | BP | Experiment |
|---|---|---|---|---|
| $O_2\cdots H_2O/O_2\cdots D_2O$ | 0.71 | 23.6 | 21.0 | 19.7[11] |
| $O_2\cdots H_2O/^{18}O_2\cdots H_2O$ | 1.04 | 1.23 | 1.20 | 1.0[37] |

The KIE is defined as the ratio between the rate constants of the systems specified in the first column.

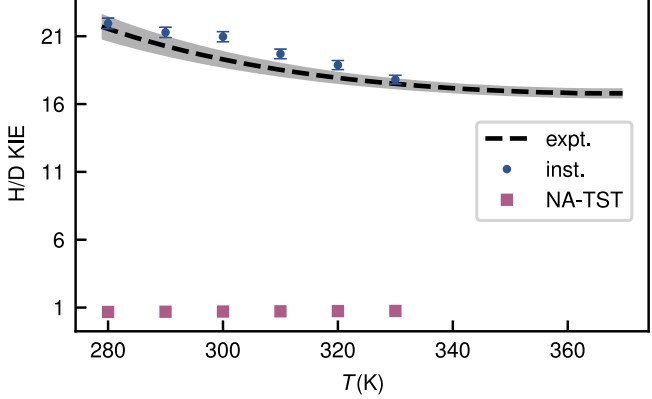

**Fig. 4 | The temperature dependence of the branch-point instanton (blue dots) H/D kinetic isotope effect (KIE), compared to experiment[11] (black dashed line) and nonadiabatic transition-state theory (NA-TST, with purple squares).** The bars on the instanton data represent its expected numerical error, as determined in Supplementary Note 2.3. The error for the experimental data is shown as a shaded grey region. Source data for the instanton results are available in Supplementary Tables 4 and 8. The fit to the experimental data (and its uncertainty) was extracted from Fig. 9 of ref. 11.

between $^{18}O_2$ and $O_2$ is much smaller than that between H and D. We assume the slightly better agreement with NA-TST here is a coincidence as it is clearly wrong for the H/D KIE. Overall, these results highlight the importance of a good description of tunnelling effects such as provided by instanton theory.

### Connection to previous theories

Ogilby and coworkers recently pointed out shortcomings of the e-to-v approach, in particular its inability to describe the temperature dependence in certain solvents[11]. They proposed a new model called 'perturbed and activated decay' (PaAD)[20,36] based on the combination of Jortner's weak-coupling theory of nonradiative processes[38] with a modified version of Marcus theory. Both of these theories are based on Fermi's golden rule. The former takes a steepest-descent approximation to evaluate the Franck–Condon overlaps within a low-temperature harmonic approximation. The latter uses a classical approximation to describe a thermally activated process. In the PaAD model, the results of these two theories are added together with prefactors determined by fitting experimental results.

Instanton theory is also based on Fermi's golden rule, but takes fewer approximations than either Jortner's or Marcus' theories. In fact, it encompasses both theories as limiting cases[39] but in general provides a unified (and explicitly temperature-dependent) first-principles formulation to span the two limits without relying on empirical parameters. For these reasons, we can assume that instanton theory recovers the results of the e-to-v and PaAD models whenever they are valid, but that in general it gives more reliable predictions than either, especially in cases where the ground and excited states exhibit significantly different frequencies and normal modes[40]. Note that as Jortner's theory assumes displaced harmonic oscillators with the same frequency in the ground and excited states, it cannot capture the branch-point singularity and the mechanism cannot be described correctly in these cases.

Our instanton calculations suggest that tunnelling plays an important role, whereas Ogilby and coworkers reported that tunnelling through the MECP is negligible[20]. However, this apparent contradiction does not necessarily invalidate the qualitative conclusions of the PaAD model. In fact, our results are in agreement with this interpretation as we also find that tunnelling does not occur through the MECP (Fig. 2b). Tunnelling is, however, strongly involved in the energy relaxation, as depicted by an instanton path which does not conserve energy (Fig. 2a). Tunnelling effects also appear implicitly in the e-to-v model as well as Jortner's theories (and by extension PaAD) as part of the quantum-mechanical Franck–Condon overlap. The instanton approach does not therefore contradict the PaAD model, but goes one step further by providing a parameter-free calculation and clear mechanistic insight for the strong isotope effects.

### Discussion

In this work, we have derived new nonadiabatic instanton methods to treat cases where the flux correlation function exhibits a branch-point singularity inside the integration contour used in standard (stationary-action) instanton theory. The branch point is caused by an infinitely large family of paths with the same tunnelling probability and is expected to be important when the curvature of the PES changes significantly between the reactant and product states. The new methods were tested on a model system of two harmonic oscillators with different frequencies and were able to accurately approximate the full quantum rate in their appropriate domain of applicability.

The branch-point singularity reveals a new tunnelling mechanism for golden-rule processes in the inverted regime. Whereas the stationary action instanton conserves energy, this is not the case in the vicinity of the singularity, and the energy of the classical trajectories on the two surfaces may differ across the nonadiabatic transition. The hopping between the PESs can therefore take place away from the

intersection seam at an energy significantly lower than that of the MECP. This is the cause of the remarkable speed-ups relative to the classical mechanism.

Using the branch-point instanton method in conjunction with on-the-fly multireference electronic structure calculations, we studied the nonradiative decay of singlet oxygen in the presence of a water molecule. In contrast to the conclusions of previous work[20], our calculations indicate the existence of significant tunnelling effects, with so much corner cutting that the mechanism becomes qualitatively different from that of the intrinsic reaction coordinate through the MECP. In particular, tunnelling occurs in the $O_2$ and O–H symmetric stretch modes. Although the tunnelling of heavy oxygen atoms has been observed previously at cryogenic temperatures[29,41], our results surprisingly show this occurring at 300 K.

The branch-point instanton method predicts rate constants that are 27 orders of magnitude faster than would be expected from a purely classical mechanism. This is an unprecedented speed-up of a chemical reaction due to quantum-mechanical effects at room temperature. Our calculations are validated by comparison with experiment. In particular, instanton theory shows excellent agreement with kinetic isotope effects observed in the experiment and correctly captures its temperature dependence.

For a more extensive test of the new theory, we should additionally consider oxygen deactivation in other solvents. For instance, in linear alkanes the rate already appears to be well described by the e-to-v model and Jortner's weak-coupling theory[11], whereas in aromatic solvents a much stronger temperature dependence is observed, such that the more flexible PaAD was so far required to fit the experimental data[20,36]. A systematic study of these cases using ab initio instanton theory is left for future work.

However, our work also has relevance far beyond that of oxygen deactivation. These new methods enable the study of nonradiative relaxation in a wide range of molecular systems. We expect to find many more examples of the importance of heavy-atom tunnelling at room temperature in solvated molecular processes relevant for chemistry and biology.

## Methods

Fermi's Golden Rule (FGR) is a quantum-mechanical method for determining the reaction rate within a perturbative approximation[22,23]. The reactant ($n = R$) and product ($n = P$) Hamiltonians are given by

$$\hat{H}_n = \sum_{j=1}^{f} \frac{p_j^2}{2m} + V_n(x), \tag{4}$$

where $x = \{x_1, ..., x_f\}$ and $p = \{p_1, ..., p_f\}$ are the positions and momenta for a set of $f$ nuclear degrees of freedom. Without loss of generality, we have assumed that all degrees of freedom have the same mass $m$. The two states are coupled by $\Delta$, which in our case is the spin−orbit coupling between the singlet and triplet states of oxygen. For simplicity, we will assume $\Delta$ to be a slowly varying function of $x$ (although this assumption can be relaxed by expanding it as a Taylor series[42,43]).

Instead of the usual FGR expression in terms of Frank−Condon overlaps, we can equivalently express the thermal rate at an inverse temperature $\beta = 1/k_B T$ in terms of a time integral over the flux correlation function[33,44,45],

$$kZ_R = \frac{|\Delta|^2}{\hbar^2} \int_{-\infty}^{\infty} dt \, c_{ff}(\tau + it), \tag{5}$$

where $Z_R = \text{Tr}[e^{-\beta \hat{H}_R}]$ is the reactant partition function and

$$c_{ff}(\tau + it) = \text{Tr}\left[ e^{-(\beta\hbar - (\tau + it))\hat{H}_R/\hbar} \, e^{-(\tau + it)\hat{H}_P/\hbar} \right]. \tag{6}$$

Note that we have made use of the slowly varying nature of $\Delta$ and taken it out of the trace.

Within certain restrictions described below, but at least for the range $\tau \in [0, \beta\hbar]$, the rate is independent of the choice of $\tau$. Common choices are $\tau = 0$ or $\tau = \beta\hbar/2$ for the standard or symmetrized correlation functions, respectively[33]. However, for now we keep it general and will discuss our choice of $\tau$ extensively later.

We use the path-integral formulation of quantum mechanics[27] to write the flux correlation function, $c_{ff}(z)$, in terms of quantum propagators $K_n(x_a, x_b, z) = \langle x_b | e^{-\hat{H}_n z/\hbar} | x_a \rangle$ which describe the evolution of the system from $x_a$ to $x_b$ in complex time $z \equiv \tau + it$ according to the Hamiltonian $\hat{H}_n$. Note that the real part of $z$ describes imaginary-time evolution (which is equivalent to real-time dynamics on the upside-down PES[46] and describes quantum Boltzmann statistics at an inverse temperature $\tau/\hbar$) while the imaginary part describes real-time evolution. This gives

$$c_{ff}(z) = \int\int dx' dx'' \, K_R(x', x'', \beta\hbar - z) K_P(x'', x', z). \tag{7}$$

We then use the semiclassical (van Vleck) approximation to the propagator[34,47,48]

$$K_n \simeq \sqrt{\frac{C_n}{(2\pi\hbar)^f}} \, e^{-S_n/\hbar}, \tag{8}$$

where the action,

$$S_n(x_a, x_b, z) = \int_0^z du \left[ \frac{1}{2} m||\dot{x}(u)||^2 + V_n(x(u)) \right], \tag{9}$$

is measured along a classical (stationary-action) trajectory from $x_a$ at $u = 0$ to $x_b$ at $u = z$ on PES $V_n$. Note that the energy,

$$E_n = -\frac{1}{2} m||\dot{x}(u)||^2 + V_n(x(u)), \tag{10}$$

is conserved along this path. Finally, $C_n$ accounts for fluctuations around the path:

$$C_n = (\pm 1)^f \left| -\frac{\partial^2 S_n}{\partial x_a \partial x_b} \right|. \tag{11}$$

Following previous work[49], we introduce a minus sign in Eq. (11) when working in the inverted regime for $C_P$ only. This ensures that both $C_R$ and $C_P$ are positive definite at the relevant values of $\tau$.

For a given value of $z$, the total action $S(X, z) = S_R(x', x'', \beta\hbar - z) + S_P(x'', x', z)$ will be stationary with respect to $X = (x' \quad x'')^T$ at some $\tilde{X}(z) = (\tilde{x}'(z) \quad \tilde{x}''(z))^T$. Integrating Eq. (7) by steepest descent[42] about this point gives

$$c_{ff}(z) \simeq \sqrt{\frac{1}{\Theta(z)}} \, e^{-\phi(z)/\hbar}, \tag{12a}$$

$$\Theta(z) = \frac{C(\tilde{X}(z), z)}{C_R(\tilde{x}'(z), \tilde{x}''(z), z) \, C_P(\tilde{x}''(z), \tilde{x}'(z), z)}, \tag{12b}$$

$$\phi(z) = S(\tilde{X}(z), z). \tag{12c}$$

Here $\phi(z)$ denotes the action along the tunnelling path which obeys the stationary-point condition for the given value of $z$. Along this path, the system evolves classically from $\tilde{x}'$ to $\tilde{x}''$ on $V_R$ in complex time $\beta\hbar - z$ and from $\tilde{x}''$ back to $\tilde{x}'$ on $V_P$ in complex time $z$. The stationary-point

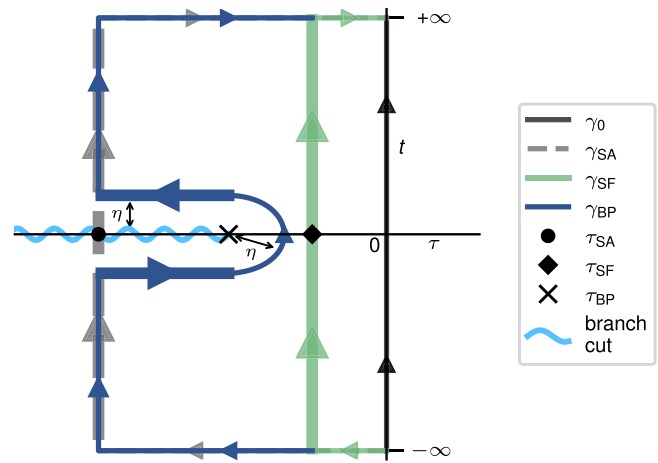

**Fig. 5 | Complex-time contours that can be used to integrate the right-hand side of Eq. (14).** The $\gamma_0$ contour is a solid black line; the stationary-action (SA) contour used in standard golden-rule instanton theory is dashed, the stationary-flux (SF) contour is in green and the branch-point (BP) contour is in dark blue. Contour segments that dominate the various integrals are drawn with thicker lines. The branch cut is represented by a wavy line.

condition ensures that the momentum is continuous at the hopping points $\tilde{x}'$ and $\tilde{x}''$[26,28]. The fluctuations around this path are accounted for by $C$, which enters in the prefactor $\sqrt{1/\Theta}$, and is defined by

$$C = (\pm 1)^f \left| \frac{\partial^2 S}{\partial X^T \partial X} \right| = (\pm 1)^f \left| \begin{matrix} \frac{\partial^2 S}{\partial x' \partial x'} & \frac{\partial^2 S}{\partial x' \partial x''} \\ \frac{\partial^2 S}{\partial x'' \partial x'} & \frac{\partial^2 S}{\partial x'' \partial x''} \end{matrix} \right|. \tag{13}$$

The minus sign is used in the inverted regime and accounts for the fact that the action is a maximum along $f$ degrees of freedom[49].

The flux correlation function is a holomorphic (complex analytic) function of the complex variable $z$, at least in some regions of the complex plane[28]. We can therefore use Cauchy's integral theorem[32], which states that the integral of a complex analytic function along a closed contour is zero, to express the time integral in Eq. (5) as the contour integral,

$$\int_{-\infty}^{\infty} \mathrm{d}t \, c_{\mathrm{ff}}(\tau + \mathrm{i}t) = -\mathrm{i} \int_{\gamma} \mathrm{d}z \, c_{\mathrm{ff}}(z). \tag{14}$$

where $\gamma$ is a contour from $z = -\mathrm{i}\infty$ to $\mathrm{i}\infty$. Note that according to this formulation, the integral along $\gamma$ will be purely imaginary, making the rate purely real. For instance, the rectangular contour $\gamma_\tau$ is defined to travel in three straight lines through the points $\{-\mathrm{i}\infty, \tau - \mathrm{i}\infty, \tau + \mathrm{i}\infty, +\mathrm{i}\infty\}$. The contour $\gamma_{\mathrm{SF}}$ shown in Fig. 5 is an example of such a contour, and the special case of $\tau = 0$ gives the contour $\gamma_0$ which recovers the original integral along the $t$-axis. The portions of the contour where $|z| \to \pm\infty$ give rise to a vanishing contribution, as $c_{\mathrm{ff}}$ goes to 0 faster than $1/|z|$ in this limit. Therefore, the only relevant part of $\gamma_\tau$ is the straight line from $\tau - \mathrm{i}\infty$ to $\tau + \mathrm{i}\infty$. The integrand along some contours can be highly oscillatory[50] and thus it is important to choose a suitable contour deformation that simplifies the evaluation of the integral.

In previous work for the normal regime, we found that the optimal contour passes through a stationary point of $\phi(z)$[26], and used it to obtain an expression for the golden-rule instanton rate. Although we subsequently extended this approach to the inverted regime[28] and employed it successfully[30], we demonstrate here that this procedure is not always sufficient as $c_{\mathrm{ff}}$ may not be complex analytic inside the optimal contour in this regime.

To guide our discussion, we consider a model system of two-dimensional harmonic oscillators

$$V_{\mathrm{R}}(x) = \frac{1}{2} m\omega_1^2 x_1^2 + \frac{1}{2} m\omega_{\mathrm{R}}^2 x_2^2, \tag{15a}$$

$$V_{\mathrm{P}}(x) = \frac{1}{2} m\omega_1^2 (x_1 - \zeta)^2 + \frac{1}{2} m\omega_{\mathrm{P}}^2 x_2^2 - \varepsilon, \tag{15b}$$

where we refer to $x_1$ as the reaction coordinate and $x_2$ as a spectator mode and all frequencies are real and positive. The problem we are investigating appears when the spectator frequencies are unequal, $\omega_{\mathrm{R}} \neq \omega_{\mathrm{P}}$. For simplicity, the two PESs have a common frequency $\omega_1$ along the reaction coordinate, although this is, in principle, not required. The parameters are chosen such that the system is in the inverted regime. This means that the gradients at the MECP are parallel leading to a sloped crossing (in contrast to the normal regime, where the gradients are antiparallel leading to a cusped barrier). As closed-form expressions for the propagators of a harmonic system are known[27], this simple model provides a useful example to illustrate the behaviour we wish to demonstrate. Note that our ab initio treatment of singlet oxygen deactivation does not rely on this simplified model.

## Stationary-action instanton theory

In standard golden-rule instanton theory[26,28,49], one first locates the stationary point of $\phi(\tau)$ (in particular a maximum), with value $\phi_{\mathrm{SA}}$, which we call the stationary action (SA) point. The deformed contour $\gamma_{\mathrm{SA}}$ corresponds to $\gamma_\tau$ with $\tau = \tau_{\mathrm{SA}}$, i.e. a straight line segment from $\tau_{\mathrm{SA}} - \mathrm{i}\infty$ to $\tau_{\mathrm{SA}} + \mathrm{i}\infty$ (see Fig. 5 and ignore the branch cut for now). The action is then approximated around $\tau_{\mathrm{SA}}$ along the $t$-axis by a second-order Taylor series, i.e. $\phi(\tau_{\mathrm{SA}} + \mathrm{i}t) \simeq \phi_{\mathrm{SA}} + \mu_{\mathrm{SA}} t^2/2$, where we use the Cauchy–Riemann equations[32] to write $\mu_{\mathrm{SA}}$ in terms of derivatives of $\tau$, i.e. $\mu_{\mathrm{SA}} = \frac{\mathrm{d}^2\phi}{\mathrm{d}t^2} = -\frac{\mathrm{d}^2\phi}{\mathrm{d}\tau^2}$ evaluated at $z = \tau_{\mathrm{SA}}$. This allows us to perform the integral by steepest descent[42], which amounts to approximating $c_{\mathrm{ff}}$ by a Gaussian centred on $\tau_{\mathrm{SA}}$,

$$c_{\mathrm{ff}}(\tau_{\mathrm{SA}} + \mathrm{i}t) \simeq c_{\mathrm{ff}}(\tau_{\mathrm{SA}}) \, e^{-\mu_{\mathrm{SA}} t^2/2\hbar}. \tag{16}$$

Here, it has been implicitly assumed that the prefactor $\sqrt{1/\Theta}$ is slowly varying in $t$. This gives us the following expression for the standard golden-rule instanton rate constant $k_{\mathrm{SA}}$:

$$k_{\mathrm{SA}} Z_{\mathrm{R}} = \frac{|\Delta|^2}{\hbar^2} \sqrt{\frac{2\pi\hbar}{\mu_{\mathrm{SA}}}} c_{\mathrm{ff}}(\tau_{\mathrm{SA}}). \tag{17}$$

The rate defined by Eq. (17) is expressed in terms of the stationary point of the action and the fluctuations around it. We shall henceforth refer to it as the stationary-action instanton. In addition to the momentum conservation, the stationary-point condition $\frac{\mathrm{d}\phi}{\mathrm{d}\tau} = 0$ enforces energy conservation between the two segments of the path on $V_{\mathrm{R}}$ and $V_{\mathrm{P}}$[26,28]. The segments join smoothly at the hopping points $\tilde{x}'$ and $\tilde{x}''$ to form a periodic orbit in imaginary time, which represents the dominant tunnelling pathway. Typically, $\tilde{x}' = \tilde{x}''$, i.e. the instanton path folds back on itself and the coupling $\Delta$ in Eq. (17) is evaluated at this point.

In the normal regime, the stationary point is located at a positive value of $\tau$[26]. The paths on $V_{\mathrm{R}}$ and $V_{\mathrm{P}}$ thus both travel in positive imaginary time, with a total period of $\beta\hbar$. In contrast, in the inverted regime, the stationary point is located at a negative value of $\tau$[28]. We have previously shown how instanton theory can be extended to such systems[28,49] by allowing the path on $V_{\mathrm{P}}$ to travel in negative imaginary time and compensating with a longer positive imaginary-time path on $V_{\mathrm{R}}$ such that total period of the instanton remains equal to $\beta\hbar$. This approach has been successfully applied to the spin-crossover reaction

of thiophosgene using ab initio electronic-structure theory to obtain excellent agreement with experiment[30].

This derivation of instanton theory relies on the assumption that $c_{ff}(z)$ is complex analytic over the region $\tau_{SA} \leq \mathrm{Re}\, z \leq 0$, such that the integral over $\gamma_{SA}$ is the same as that over $\gamma_0$. For our problem, the exponential term $\exp(-\phi/\hbar)$ is always well-behaved, finite, and non-zero. However, we have observed that for a certain range of parameters for the model system in the inverted regime, $\Theta(z)$ has a root on the negative $\tau$-axis, making $c_{ff}(z)$ singular at this point. Specifically, it is $C(\tau)$ that has a root at a non-zero $\tau$. Note that although $C_R$, $C_P$ and $C$ are singular at $z = 0$, the singularities cancel exactly such that $\sqrt{1/\Theta}$ is analytic at this point. In particular, using explicit expressions for the model system (provided in Supplementary Note 1),

$$C(\tau) = m^2 \omega_1^2 \mathcal{T}_1(\tau)\, m^2 \omega_R \omega_P \mathcal{T}_2(\tau), \tag{18a}$$

$$\mathcal{T}_1(\tau) = \left[1 + \frac{\tanh\left(\frac{\omega_1 \tau_R}{2}\right)}{\tanh\left(\frac{\omega_1 \tau_P}{2}\right)}\right]\left[1 + \frac{\tanh\left(\frac{\omega_1 \tau_P}{2}\right)}{\tanh\left(\frac{\omega_1 \tau_R}{2}\right)}\right], \tag{18b}$$

$$\mathcal{T}_2(\tau) = \frac{\omega_P}{\omega_R}\left[\frac{\omega_R}{\omega_P} + \frac{\tanh\left(\frac{\omega_R \tau_R}{2}\right)}{\tanh\left(\frac{\omega_R \tau_P}{2}\right)}\right]\left[\frac{\omega_R}{\omega_P} + \frac{\tanh\left(\frac{\omega_P \tau_P}{2}\right)}{\tanh\left(\frac{\omega_P \tau_R}{2}\right)}\right], \tag{18c}$$

where we have defined $\tau_R \equiv \beta\hbar - \tau$ and $\tau_P \equiv \tau$ as the imaginary time spent on $V_R$ and $V_P$, respectively.

For $0 < \tau < \beta\hbar$, which is relevant for the normal regime, both $\mathcal{T}_1$ and $\mathcal{T}_2$ are positive definite, so no singularities can occur. However, for $\tau < 0$, which is relevant for the inverted regime, $\mathcal{T}_1$ is negative definite, but $\mathcal{T}_2$ can cross zero and change sign. Assuming that the root is not also a stationary point, $C(\tau)$ is linear in the displacement from the zero of $\mathcal{T}_2$ to leading order and we find that $c_{ff}(\tau)$ has an inverse square-root singularity. The location of the zero of $C(\tau)$ is denoted $\tau_{BP}$ as it is a branch point (BP) of $c_{ff}(z)$. When $\tau > \tau_{BP}$, $\Theta(\tau)$ is positive and hence $\sqrt{1/\Theta(\tau)}$ is purely real, whereas for $\tau < \tau_{BP}$, $\Theta(\tau)$ is negative and $\sqrt{1/\Theta(\tau)}$ is purely imaginary.

If this singularity is inside the region bounded by the original and the deformed contours, $\gamma_0$ and $\gamma_{SA}$, i.e. $\tau_{SA} < \tau_{BP}$ (see Fig. 5), Cauchy's integral theorem no longer applies. Note that the singularity is not a pole, so one cannot apply the residue theorem either. This means that the rate given by the stationary-action instanton [Eq. (17)] is invalid. More generally, this means that the rate expressed by Eq. (5) is only independent of the choice of $\tau$ as long as $\tau > \tau_{BP}$.

Note that $\tau_{BP} \to -\infty$ as $\omega_P/\omega_R \to 1$, which is why this behaviour was not observed in our previous studies of the spin-boson model[28,39,40]. To explore this problem further, it helps to consider three different regions, in which the relative values of $\tau_{BP}$ and $\tau_{SA}$ are such that: (I) the singularity is far to the left of the stationary-action time, $\tau_{BP} \ll \tau_{SA}$; (II) the two times are nearly coincident, $\tau_{BP} \approx \tau_{SA}$; (III) the singularity is far to the right of the stationary-action time, $\tau_{BP} \gg \tau_{SA}$, and lies within the stationary-action contour. Region I can already be tackled using standard instanton theory [Eq. (17)]. In order to derive semiclassical theories for the other two regions, we introduce new contour deformations in the following sections and use the model system to benchmark the results.

### Stationary-flux instanton theory

One approach, formally applicable in all three regions is to locate a stationary point of $c_{ff}$ (rather than of $\phi$), which we name $\tau_{SF}$, the stationary-flux (SF) point. As $c_{ff}(\tau)$ is infinite at $\tau_{BP}$, a minimum always exists at some $\tau_{SF} > \tau_{BP}$. Cauchy's integral theorem can therefore be used to evaluate the rate along the contour $\gamma_{SF}$ which passes through $\tau_{SF}$, as illustrated in Fig. 5.

The procedure to calculate the semiclassical rate using steepest-descent integration along $\gamma_{SF}$ is quite straightforward as it is very

similar to that of the stationary-action instanton method. Again $c_{ff}$ is approximated as a Gaussian,

$$c_{ff}(\tau_{SF} + it) \approx c_{ff}(\tau_{SF})\, e^{-\mu_{SF} t^2/2\hbar}, \tag{19}$$

where

$$\mu_{SF} = \hbar \frac{d^2 \ln c_{ff}}{d\tau^2}\bigg|_{\tau = \tau_{SF}}. \tag{20}$$

In contrast to the stationary-action method, here we have not made the assumption that the prefactor in (12) is slowly varying and it instead is included in the steepest-descent procedure. This is crucial in order to capture the effect of the rapidly varying prefactor near the singularity. However unlike Eq. (16), Eq. (19) is formally not an asymptotic approximation to $c_{ff}$ in $\hbar$, although as we will see, it may still behave quite accurately. This results in the following approximation to the golden-rule rate:

$$k_{SF} Z_R = \frac{|\Delta|^2}{\hbar^2}\sqrt{\frac{2\pi\hbar}{\mu_{SF}}}\, c_{ff}(\tau_{SF}). \tag{21}$$

This approach is similar in spirit to the quantum instanton[51–53] and Wolynes theory[44,54,55], both of which locate stationary points of the correlation function. The key difference is that we carry out a steepest-descent integration over the position coordinates in addition to the time variable. This makes the overall calculation far more efficient but, like all semiclassical instanton methods, means that we cannot apply it to the liquid phase. An important advantage is that just like the stationary-action instanton, it can be applied directly to the inverted regime[28], whereas Wolynes theory requires extrapolation[56].

Note that although an analytic expression for $\mu_{SF}$ can be obtained for the model system, in general one would require third and fourth derivatives of the PESs. In order to avoid this potentially expensive calculation, the derivatives in Eq. (20) were evaluated numerically using information from stationary-action paths with fixed values of imaginary time $\tau$, which we call fixed-$\tau$ instantons. Further details of the evaluation of $\mu_{SF}$ are given in Supplementary Note 2.3.

The individual trajectories on $V_R$ and $V_P$ are classical and the stationary-point condition $\frac{\partial S}{\partial X} = 0$ ensures that the momentum is conserved at the hopping point. However unlike the stationary-action instanton, where the condition $\frac{d\phi}{d\tau} = E_P - E_R = 0$ enforces energy conservation between the two segments, the stationary-flux instanton does not conserve energy as defined in Eq. (10). However, the stationary-flux condition $\frac{dc_{ff}}{d\tau} = 0$ is equivalent to the conservation of the energy expectation value over the density matrix of the instanton, i.e. $\mathrm{Tr}[\hat{H}_R \hat{\rho}(\tau_{SF})] = \mathrm{Tr}[\hat{H}_P \hat{\rho}(\tau_{SF})]$ with $\hat{\rho}(\tau) = e^{-(\beta\hbar - \tau)\hat{H}_R/\hbar}\, e^{-\tau\hat{H}_P/\hbar}$. The stationary-flux instanton can be interpreted as conserving the quantum energy (including zero-point energy) rather than just the classical energy [Eq. (10)], which is the case for the stationary-action instanton.

### Branch-point instanton theory

A more rigorous solution to our problem involves treating the singularity, which is a branch-point of a square-root function, explicitly. As we shall see, this approach has the advantage that the approximate $c_{ff}$ is expressed by an asymptotic relation and is thus exact in the $\hbar \to 0$ limit.

Similar to the stationary-flux method, we require a contour that avoids the branch point and does not cross the branch cut so as to not violate Cauchy's integral theorem. We choose a contour $\gamma_{BP}$ (Fig. 5) such that it follows $\gamma_{SA}$ until a small distance $\eta$ before the branch cut. It then runs parallel to the branch cut until it reaches the branch point $\tau_{BP}$, crosses to the other side of the branch cut following a semicircular

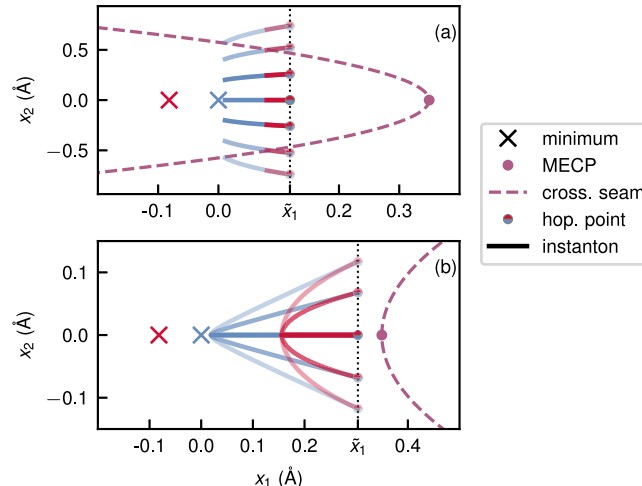

**Fig. 6 | The family of branch-point instantons of the model system at 300 K.** The model system is described by Eq. (15) with $\omega_1 = 1000$ cm$^{-1}$, $\zeta = -0.08$ Å, $m = 8.0$ u, $\varepsilon = 0.0345$ E$_h$ and (a) $\omega_R = 240$ cm$^{-1}$, $\omega_P = 480$ cm$^{-1}$ and (b) $\omega_R = 960$ cm$^{-1}$, $\omega_P = 240$ cm$^{-1}$. Blue corresponds to the reactant and red to the product. Transparency is added to the instantons for illustration purposes.

arc of radius $\eta$ and is completed by mirroring the path below the branch cut.

As before, the segments of the contour at $t = \pm\infty$ give rise to vanishing contributions to the integral. In the limiting case of a small $\eta$, the integral along the semicircular arc also vanishes. If we assume that $\tau_{BP} \gg \tau_{SA}$ (region III), the integral along the contour will be dominated by the neighbourhood of the branch point. Therefore $c_{ff}(\tau)$ will have decayed to a negligibly small value at $\tau_{SA}$ making the vertical segments in the $t$-direction subdominant. This leads to

$$\int_{\gamma_{BP}} dz\, c_{ff}(z) \simeq \lim_{\eta \to 0^+} 2i\, \text{Im}\left[\int_{\tau_{SA}}^{\tau_{BP}} d\tau\, c_{ff}(\tau - i\eta)\right], \quad (22)$$

where we have used the fact that the real parts of a function on either side of a square-root branch cut are equal while the imaginary parts have equal magnitude but opposite sign.

We then approximate the integral in Eq. (22) by steepest descent[42]. Note that this procedure differs from that of the stationary-action and stationary-flux instanton theories, as here the maximum of the integrand occurs at the edge of the integration range (at $\tau_{BP}$) and is not a stationary point. For this reason, it is only necessary to expand $\phi$ to first order in $\tau$ around $\tau_{BP}$. In order to account for the singularity in the prefactor, we must additionally expand $\Theta(\tau)$ to leading order in the displacement from $\tau_{BP}$. This results in the following asymptotic approximation,

$$c_{ff}(\tau - i\eta) \simeq \sqrt{\frac{1}{(\tau - i\eta - \tau_{BP})\Omega_{BP}}} e^{-(\phi_{BP} - (\tau - i\eta - \tau_{BP})\mathcal{E}_{BP})/\hbar}, \quad (23)$$

where $\phi_{BP} = \phi(\tau_{BP})$ and

$$\Omega_{BP} = \frac{d\Theta}{d\tau}\bigg|_{\tau = \tau_{BP}}, \quad (24a)$$

$$\mathcal{E}_{BP} = -\frac{d\phi}{d\tau}\bigg|_{\tau = \tau_{BP}}. \quad (24b)$$

Note that for $\tau > \tau_{SA}$, $\phi(\tau)$ is a monotonically decreasing function and $\mathcal{E}_{BP}$ is therefore positive in this region. Similarly, we know that the prefactor term $\Theta$ is always positive for $0 < \tau < \beta\hbar$ and therefore its slope at the first root on the negative $\tau$-axis, $\Omega_{BP}$, must also be positive.

If $\tau_{BP}$ and $\tau_{SA}$ are well separated, the integral is dominated by the region around $\tau_{BP}$ and we can change the lower limit of integration in Eq. (22) from $\tau_{SA}$ to $-\infty$. Despite the presence of the singularity in the limit $\eta \to 0^+$, the resulting integral can be evaluated analytically. This results in an asymptotic approximation to the rate constant that only uses information obtained at the branch point

$$k_{BP}Z_R = \frac{|\Delta|^2}{\hbar^2} \sqrt{\frac{4\pi\hbar}{\Omega_{BP}\mathcal{E}_{BP}}} e^{-\phi_{BP}/\hbar}. \quad (25)$$

In the standard case (region I), the instanton and small fluctuations around it dominate the tunnelling mechanism due to the stationary-action principle. However, in region III, the mechanism is completely different due to the existence of the branch-point singularity. When one encounters an infinity in semiclassical theories, it is normally a sign that an approximation has been made which is not valid. However, this is not the case here as integration around the branch point leads to a finite rate constant, as required. The infinity in $c_{ff}(z)$ is a result of a zero mode in path space, along which the change of action on one surface is perfectly cancelled out by the change of action on the other. The branch point can therefore be considered to originate from an extreme entropic effect in which there is an infinite family of paths which contribute equally.

As illustrated in Fig. 6 for the model system, the family of branch-point instantons all have hopping points with the same $x_1$ coordinate, but can take any value of $x_2$ such that $x_2'' = +x_2'$ when $\omega_P > \omega_R$ or $x_2'' = -x_2'$ when $\omega_R > \omega_P$. Therefore, depending on the regime, the eigenvector of the zero mode will correspond to moving both hopping points in either a symmetric or an antisymmetric fashion.

Similar to the stationary-flux instanton, the segments of the branch-point instantons are classical trajectories and although the classical energy is not conserved, the momentum remains continuous throughout the trajectory. However, unlike the stationary-flux instanton, the energy difference of the dominant tunnelling mechanism is not uniquely defined, as every path in the ensemble contributes equally to the reaction mechanism and can have a different energy gap between the reactant and product segments.

When we discuss the mechanistic insight obtained from the branch-point instanton in the 'Results' section, we will use the central path of the family, defined as the limit of the fixed-$\tau$ optimisation procedure as $\tau \to \tau_{BP}$. This is qualitatively similar to that of the stationary-flux instanton.

**Uniform approximation**

In region II ($\tau_{BP} \approx \tau_{SA}$), neither the branch-point [Eq. (25)] nor the stationary-action approach [Eq. (17)] is valid and both diverge when $\tau_{SA} = \tau_{BP}$. In the former case, this is because $\mathcal{E}_{BP}$ approaches 0, while in the latter case, it is because $\Theta$ approaches 0. This can be resolved with a uniform asymptotic approximation to $c_{ff}$ that is valid across all three regions.

Let us first consider the case where $\tau_{BP} > \tau_{SA}$, for which we define the contour $\gamma_U$ such that it is coincident with $\gamma_{BP}$. When $\tau_{BP}$ approaches $\tau_{SA}$, we can no longer make the approximation that $c_{ff}(\tau - i\eta)$ is sufficiently decayed at $\tau_{SA}$ and hence cannot shift the lower integration bound in Eq. (22) to $-\infty$. Additionally, we must also retain the integral along the $t$-direction, giving

$$\int_{\gamma_U} dz\, c_{ff}(z) \simeq 2i \lim_{\eta \to 0^+} \left( \text{Im}\left[\int_{\tau_{SA}}^{\tau_{BP}} d\tau\, c_{ff}(\tau - i\eta)\right] + \text{Re}\left[\int_{-\infty}^{-\eta} dt\, c_{ff}(\tau_{SA} + it)\right] \right). \quad (26)$$

The first integral (along the branch cut) is treated similarly to the branch-point method; the only difference is that we keep the lower

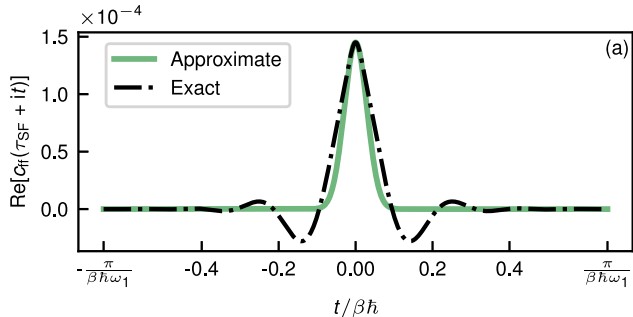

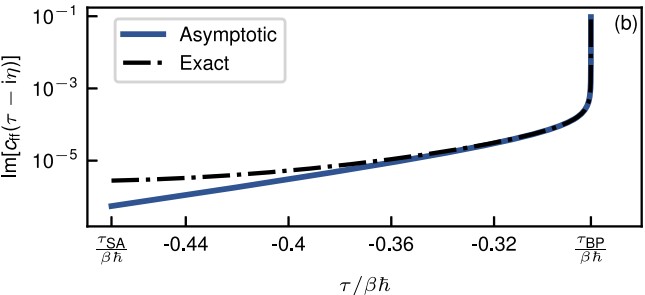

**Fig. 7 | A comparison between the exact and approximate forms of the flux correlation function for the model system.** The functions are shown along the dominant segment of **a** $\gamma_{SF}$ and **b** $\gamma_{BP}$. For the latter, a small value of $\eta/\beta\hbar = 9.5 \times 10^{-9}$ was chosen.

bound at $\tau_{SA}$. The second integral (in the $t$-direction) can be evaluated using a combination of the methods used to derive the stationary-action and branch-point instanton theories. Expanding $\phi$ and $\Theta$ around $\tau_{SA}$ to leading order in $t$ results in the asymptotic approximation

$$c_{ff}(\tau_{SA} + it) \simeq \sqrt{\frac{1}{\Theta_{SA} + i\Omega_{SA}t}} e^{-(\phi_{SA} + \mu_{SA}t^2/2)/\hbar}, \qquad (27)$$

where $\Theta_{SA} = \Theta(\tau_{SA})$ and $\Omega_{SA} = \frac{d\Theta}{d\tau}\big|_{\tau_{SA}}$. Therefore, with the asymptotic form of $c_{ff}(\tau - i\eta)$ from Eq. (23), we get a uniform asymptotic approximation to $c_{ff}$ and the corresponding rate constant is

$$k_U = k_{BP}\,\mathrm{Erf}\left(\sqrt{\frac{\mathcal{E}_{BP}(\tau_{BP} - \tau_{SA})}{\hbar}}\right) + k_{SAU}, \qquad (28)$$

where the contribution to the uniform approximation from the stationary-action contour, $k_{SAU}$, is a simple one-dimensional integral

$$k_{SAU}Z_R = \frac{|\Delta|^2}{\hbar^2} e^{-\phi_{SA}/\hbar} \int_{-\infty}^{+\infty} dt\, \sqrt{\frac{1}{\Theta_{SA} + i\Omega_{SA}t}} e^{-\mu_{SA}t^2/2\hbar}, \qquad (29)$$

that is easily evaluated numerically. Note that for the $k_{SAU}$ term [Eq. (29)], the coupling $\Delta$ is evaluated at the hopping point corresponding to $\tau_{SA}$, whereas for the $k_{BP}$ term [Eq. (25)], it is evaluated at the hopping point corresponding to $\tau_{BP}$.

When $\tau_{BP} \leq \tau_{SA}$, we instead define $\gamma_U$ such that it is coincident with $\gamma_{SA}$. The dominant contribution to the integral still comes from $\tau_{SA}$ but as $\Theta$ approaches 0 in this region, the next-order $t$-dependent term of $\Theta$ must also be taken into account. The correct asymptotic approximation to $c_{ff}$ is thus given by Eq. (27) and $k_U = k_{SAU}$ in this case.

In this way, $k_U$ is valid in all three regions and (unlike $k_{SF}$) is a rigorous asymptotic approximation that becomes exact in the $\hbar \to 0$ limit.

## Model system benchmark

In order to test the accuracy of the approximations made, we first present $c_{ff}$ [Eq. (12)] for the model system described in Eq. (15) with parameters $\omega_1 = 1000\ \mathrm{cm}^{-1}$, $\zeta = -0.08\ \text{Å}$, $m = 8.0\ \mathrm{u}$ and $\varepsilon = 0.0345\ \mathrm{E_h}$ at $T = 300\ \mathrm{K}$. Unless otherwise specified, $\omega_R = 240\ \mathrm{cm}^{-1}$ and $\omega_P = 480\ \mathrm{cm}^{-1}$. The parameters were chosen in such a way that they roughly correspond to a two-dimensional subspace of the $O_2\cdots H_2O$ system. In particular, $\omega_R$ and $\omega_P$ correspond to a representative spectator mode, the bias $\varepsilon$ accounts for the potential energy difference between the singlet and triplet minima and the shift $\zeta$ is chosen such that the energy at the MECP is $0.066\ \mathrm{E_h}$. Note that we only use this model system to demonstrate the efficacy of the new methods and we will not use it to make predictions about the rate and mechanism for the decay of singlet oxygen.

In Fig. 7, we compare the accuracy of the approximations to $c_{ff}$ along the dominant segments of the contours $\gamma_{SF}$ and $\gamma_{BP}$. In the case of $\gamma_{SF}$, while the approximate form is accurate near the peak, it does not account for the oscillations. This inaccuracy is a consequence of the fact that Eq. (19) is not a true asymptotic approximation to $c_{ff}$ around $\tau_{SF}$. In the case of $\gamma_{BP}$, the asymptotic form is a very good approximation to the exact function in the dominant region around $\tau_{BP}$, with a visible deviation only in a subdominant region that does not contribute significantly to the integral.

Note that for the simple model system, an exact quantum rate constant is not strictly defined. This is because such a bound, quantised system with only two degrees of freedom tends to exhibit coherent dynamics, resulting in recurrences in $c_{ff}$ at large values of $t$. However, one can still define a short-time version of the rate constant by choosing a finite range for the integration over $t$, such that the flux correlation function has decayed to a small value at the boundary of the range. This truncation is equivalent to smearing the vibrational states[49]. We have thus chosen to define the quantum-mechanical rate constant by integrating $c_{ff}$ along $\gamma_{SF}$ from $\tau_{SF} - i\pi/\omega_1$ to $\tau_{SF} + i\pi/\omega_1$. As illustrated in Fig. 7a, the short-time peak is sufficiently decayed by this point. For a realistic system with several modes for energy dissipation, these recurrences do not occur, which justifies the procedure.

Before we discuss the rates, we present the dependence of the dominant imaginary times across a range of $\omega_P$ with a fixed value of $\omega_R = 240\ \mathrm{cm}^{-1}$ in Fig. 8a. For the model system, the tunnelling path lies completely along the reaction mode $x_1$ and $\tau_{SA}/\beta\hbar = -0.469$ is independent of $\omega_R$ and $\omega_P$. As $\omega_P/\omega_R \to 1$, the system approaches a two-dimensional spin-boson model and the branch point is pushed to $-\infty$ meaning that the system will be in region I regardless of the other parameters. In contrast, the branch point $\tau_{BP}$ approaches 0 as the ratio $\omega_P/\omega_R$ increases and has a finite, non-zero limit as the ratio tends to 0. For the chosen parameters, this limit is more negative than $\tau_{SA}$ and so a second pair of regions II and III do not appear on the left-hand side of the plot. However this is not always the case, as in different parameter regimes, it is possible to find $\tau_{SA} < \tau_{BP}$ when $\omega_P/\omega_R < 1$. The branch point is thus expected to play a role in the deep inverted regime (where $\tau_{SA}$ becomes more negative) or strongly asymmetric systems, especially in the $\omega_P \gg \omega_R$ case.

In Fig. 8b, we compare the various instanton rates to the quantum result. In region I, we find that $k_{SA}$, $k_{SF}$ and $k_U$ are nearly identical and in good agreement with the full quantum results. This is because the prefactor term $\Theta$ in Eq. (12) is well approximated by a constant in this region and the exponential term $e^{-\phi(z)/\hbar}$ is well approximated by a Gaussian.

In region III, as expected, the rates calculated using $k_{BP}$ and $k_U$ are very similar and have an accuracy comparable to that of $k_{SA}$ in region I. On the other hand, $k_{SF}$ is less accurate because unlike in region I, where $\tau_{SF} \approx \tau_{SA}$, $c_{ff}(\tau_{SF} + it)$ is not always well approximated by a Gaussian in region III, as illustrated in Fig. 7a. As discussed above, $k_{SF}$ is not a true asymptotic approximation to the quantum rate because it does not

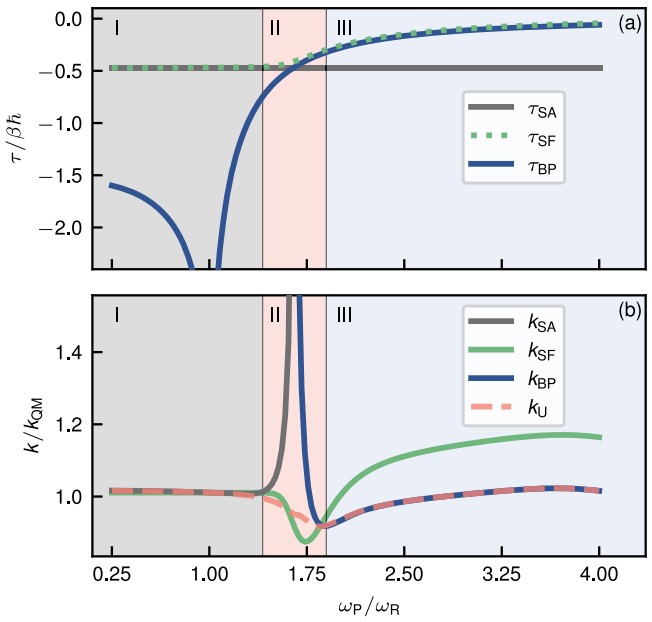

**Fig. 8 | Benchmark data for the model system. a** The dominant imaginary times $\tau_{SA}$ (stationary-action point), $\tau_{SF}$ (stationary-flux point) and $\tau_{BP}$ (branch point). **b** The ratio of the instanton rate constants [Eqs. (17), (21), (25) and (28)] to the exact quantum rate constant (obtained by numerical integration along the short-time part of $\gamma_{SF}$). Both (**a**) and (**b**) are plotted as a function of the ratio of the spectator-mode frequencies. The background colours identify the three regions, defined as I: $\tau_{BP} \ll \tau_{SA}$, II: $\tau_{BP} \approx \tau_{SA}$ and III: $\tau_{BP} \gg \tau_{SA}$.

recover the correct quantum result in the limit $\hbar \to 0$. Despite this, it should be noted that the relative error of $k_{SF}$ is still quite modest, remaining below 20%.

In region II, both $k_{BP}$ and $k_{SA}$ are poor approximations to $c_{ff}$ and the calculated rates diverge significantly from the quantum rate. While $k_{SF}$ does not diverge in this region, it is less accurate than it is in region I. It is to fix this that the uniform approximation was developed, and we see that it smoothly connects $k_{BP}$ in region III to $k_{SA}$ in region I. The value of the error function in Eq. (28) gives a simple and reliable measure of whether it is necessary to use $k_U$ or whether $k_{BP}$ is sufficient; if the error function is close to 1, $\tau_{SA}$ and $\tau_{BP}$ are well separated and it is unnecessary to compute $k_{SAU}$.

### Ring-polymer formulation

We have derived new semiclassical approximations to the FGR rate and presented some benchmarks for a model harmonic system. In order to evaluate these rates for anharmonic systems where analytic expressions for the action are not known, we use the ring-polymer formalism[31,34,49]. In this formulation, the path is discretized into $N$ beads $\mathbf{r} = \{\mathbf{r}_1, \mathbf{r}_2, ..., \mathbf{r}_N\}$, where $\mathbf{r}_i$ is an $f$-dimensional vector that corresponds to the configuration of the molecule at bead $i$. For a molecule with $N_{atom}$ atoms and $f = 3N_{atom}$ degrees of freedom, the discretized action is

$$
\begin{aligned}
S_N(\mathbf{r}, \tau) = & \sum_{i=1}^{N_R} \sum_{a=1}^{N_{atom}} \left( \frac{m_a ||\mathbf{r}_{i,a} - \mathbf{r}_{i-1,a}||^2}{2\epsilon_R} + \frac{1}{2}\epsilon_R [V_R(\mathbf{r}_i) + V_R(\mathbf{r}_{i-1})] \right) \\
& + \sum_{i=N_R+1}^{N} \sum_{a=1}^{N_{atom}} \left( \frac{m_a ||\mathbf{r}_{i,a} - \mathbf{r}_{i-1,a}||^2}{2\epsilon_P} + \frac{1}{2}\epsilon_P [V_P(\mathbf{r}_i) + V_P(\mathbf{r}_{i-1})] \right),
\end{aligned}
$$
(30)

where the reactant trajectory is split into $N_R$ intervals of length $\epsilon_R = \tau_R/N_R$ and the product into $N_P = N - N_R$ intervals of length $\epsilon_P = \tau_P/N_P$, $m_a$ is the mass of atom $a$, and $\mathbf{r}_{i,a}$ is the three-dimensional position vector of atom

$a$ in bead $i$. The index $i$ is cyclic such that $\mathbf{r}_N \equiv \mathbf{r}_0$. Note that in the inverted regime, $\epsilon_P$ is negative.

The corresponding expression for the flux correlation function, valid for negative $\tau$, is

$$
\begin{aligned}
c_{ff}(\tau) = & \left( \frac{1}{i^{N_P f}} \right) \prod_{a=1}^{N_{atom}} \left( \frac{m_a}{2\pi\hbar\epsilon_R} \right)^{3N_R/2} \left( \frac{m_a}{2\pi\hbar|\epsilon_P|} \right)^{3N_P/2} \\
& \times \int d\mathbf{r}\, e^{-S_N/\hbar},
\end{aligned}
$$
(31)

where the extra factors of i arise from the integration along the complex position axes in the inverted regime[49].

The integral over $\mathbf{r}$ in Eq. (31) is performed by steepest descent[42] around a point $\tilde{\mathbf{r}}$ that makes $S_N$ stationary. The $f_0$ modes that correspond to translations and rotations of the ring polymer are excluded from the product and are instead treated with the expressions for the translational and rotational partition functions of a ring polymer, $Z_{inst}^{trans}$ and $Z_{inst}^{rot}$[34,49]. The $N$-bead discretized version of $c_{ff}$ is therefore

$$
c_{ff} = Z_{inst}^{trans} Z_{inst}^{rot} \sqrt{\frac{1}{\Theta}} e^{-\phi/\hbar},
$$
(32)

where

$$
\phi(\tau) = S_N(\tilde{\mathbf{r}}(\tau), \tau).
$$
(33)

Note that the optimal bead positions $\tilde{\mathbf{r}}$ depend on $\tau$.

The prefactor $\sqrt{1/\Theta}$ contains information about the index and curvature of the stationary point. In the normal regime, this stationary point is a minimum, while in the inverted regime, it is a saddle point of index $K$. As long as $\tau > \tau_{BP}$, the index is $K = N_P f$[28]. This requires changing the integration contour to point along imaginary position axes for $N_P f$ degrees of freedom. The resulting factor of $i^{N_P f}$ cancels exactly with the one from Eq. (31). However as $\tau$ crosses $\tau_{BP}$, one of the eigenvalues of the Hessian of $S_N$ changes sign and consequently the saddle point has an index $K = N_P f - 1$. In general, there may be multiple branch points and the index $K$ reduces by 1 across each one. The discretized version of $\Theta$, valid for general $K$, is

$$
\Theta(\tau) = (-1)^{K-N_P f} \frac{\left[ (\beta_N \hbar)^N \epsilon_R^{N_R} |\epsilon_P|^{N_P} \right]^f}{(\beta\hbar)^{2f_0}} \prod_{j=f_0+1}^{Nf} |\lambda_j|,
$$
(34)

where $\beta_N = \beta/N$ and $\lambda_j$ are the eigenvalues of the mass-weighted Hessian of $S_N/\beta_N\hbar$ with respect to $\mathbf{r}$. These quantities are computed from fixed-$\tau$ instanton optimisations at a few selected values of $\tau$. The derivatives $\mu_{SF}$ [Eq. (20)], $\Omega_{BP}$ [Eq. (24a)] and $\mathcal{E}_{BP}$ [Eq. (24b)] are computed numerically by fitting $\ln c_{ff}(\tau)$, $\Theta(\tau)$ and $\phi(\tau)$ with splines.

### Electronic structure

The ab initio PESs for the singlet ($V_R$) and triplet ($V_P$) spin multiplicities were evaluated separately using state-specific multiconfigurational self-consistent field MCSCF(12,8), with active orbitals that consist of the valence molecular orbitals localised on $O_2$. Unless otherwise reported, all calculations were performed in vacuum and with the cc-pVDZ basis set. Analytic MCSCF gradients and Hessians were used to evaluate the derivatives of $V_R$ and $V_P$. GAMESS-US[57,58] was used for all electronic-structure calculations.

To estimate $K_c$, MCSCF/cc-pVDZ with an equivalent (12,8) active space was used for isolated $O_2$. As the optimised active orbitals of $O_2 \cdots H_2O$ in its minimum geometry were localised on $O_2$, Hartree–Fock/cc-pVDZ was used for isolated $H_2O$.

In the SI, we present data for $O_2 \cdots H_2O$ calculations performed with the cc-pVTZ basis set, which resulted in $k_{BP} = 34.6$ ms$^{-1}$, i.e. a 60%

increase. While this result is expected to be more accurate, it is of the same order of magnitude as that with the cc-pVDZ basis set and does not qualitatively change our conclusions.

Adding a polarisable continuum solvation model (PCM) results in almost no change to $k_{BP}$ (19.1 ms$^{-1}$) or the spin−orbit coupling at the hopping bead of the instanton (35.95 cm$^{-1}$, as opposed to 35.61 cm$^{-1}$ for the instanton without PCM). From the various types of solvent−$O_2$ interactions proposed by Ogilby and coworkers[17], our results indicate that at least for water, the dominant contributions to the nonradiative decay rate come from short-range solvent−$O_2$ interactions and not the bulk dielectric effects of the solvent.

The spin−orbit coupling was calculated as the off-diagonal term of the full Breit−Pauli Hamiltonian, in the basis of the singlet and triplet MCSCF states (SO-CASCI)[59]. Using SO-MCQDPT[60], which additionally includes electron correlation, results in a small 4% increase to the spin−orbit coupling.

## Numerical optimisation and spline fit

Half-ring polymers were used for the fixed-$\tau$ instanton optimisation[49] to reduce the computational cost. After the optimisation was completed, the full ring polymer was created and used to evaluate the fluctuation factors for the subsequent calculations. Calculations were performed with both $N = 256$ and $N = 512$ beads, which was found to give sufficient convergence.

In order to locate the branch points, fixed-$\tau$ ring-polymer optimisations were performed at a range of $\tau$ values and splines were fitted to the calculated $\Theta(\tau)$ to locate its root. An additional fixed-$\tau$ instanton calculation was performed at the root of the spline and tested for convergence: if the relative difference between the new and old roots was less than 1%, it was chosen as the branch point $\tau_{BP}$. The derivative of the spline was used to calculate $\Omega_{BP}$. A separate spline was fitted to $\phi(\tau)$ over the same $\tau$ values to calculate $\mathcal{E}_{BP}$ from $\phi(\tau)$. A similar spline-fitting procedure was employed to find the stationary-flux point and its second derivative $\mu_{SF}$ from $\ln c_{ff}(\tau)$. Quintic splines were used in all cases. The range of $\tau$ was chosen such that it included the branch point or stationary point and data was gathered as close as possible to the neighbourhood of these key points. An example of the fitting procedure is shown in Supplementary Fig. 4 and Supplementary Fig. 5.

The MECP was obtained by optimising an effective potential $V_{eff} = (1 - \lambda)V_R + \lambda V_P$, where $\lambda$ is a Lagrange multiplier. An optimisation tolerance of $10^{-4}$ E$_h$ Å$^{-1}$ was used for locating the stationary points of $V_R$, $V_P$, $V_{eff}$ (minima, saddle points, MECPs) and $S_N/\beta_N \hbar$ (ring polymer) of the appropriate index.

## Data availability

The data generated in this study are provided in the Supplementary Information and Source Data files (Supplementary Data 1).

## Code availability

The code used to generate data presented in this study is available upon request from the corresponding author.

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

## Acknowledgements

The authors thank Frank Jensen for introducing them to the oxygen deactivation problem and Joseph Lawrence for insightful comments and discussions. This work is financially supported by the Swiss National Science Foundation through SNSF Project No. 207772, received by I.M.A., G.T. and J.O.R.

## Author contributions

G.T. and J.O.R. designed research; I.M.A. and E.R.H. performed research; I.M.A. analysed data; I.M.A., E.R.H., G.T. and J.O.R. wrote the paper.

## Competing interests

The authors declare no competing interests.
