## [Peer Review File · Nature Communications]

Heavy-atom tunnelling in singlet oxygen deactivation predicted by instanton theory with branch-point singularitiesReviewer #1 (Remarks to the Author):

This paper addresses an important topic, and the authors present their theoretical and computational work in the context of a long-standing important and challenging problem: the solvent dependent nonradiative deactivation of singlet oxygen.

However, in their presentation, I believe the authors put "the cart before the horse" by providing an extensive preface on singlet oxygen, leading the reader to infer that their work resolves the singlet oxygen deactivation problem. This sentiment is clearly expressed in the abstract where the authors indicate that their approach "holds the key to elucidating the reaction mechanism."

The authors indicate that the impetus for their study is embodied in two papers (their references 11 and 20) in which the possible contribution of tunneling in the deactivation of singlet oxygen was both experimentally and computationally considered. Outlined below are detailed comments on these published papers, as they apply to the present submission to Nature Communications. Of note is the fact that (a) evidence against a tunneling contribution was eventually provided, and (b) a satisfactory computational model, so-called PaAD, that does not consider tunneling, accounts for all the temperature dependent experimental data.

Despite the rather extensive work from Richardson's group reported in this manuscript, they do not satisfactorily "solve" the singlet oxygen problem, nor do they provide substantive evidence that argues against the results of the PaAD model described in their references 20 and 35.

Rather, Richardson and colleagues should re-cast their presentation to say that they have developed an approach to deal with tunneling through barriers in the Marcus inverted region, which can be relevant for a host of reactions, including the solvent-mediated nonradiative deactivation of singlet oxygen. They should then mention that, as noted in references 11, 20 and 35, to properly address the singlet oxygen problem, their computations must eventually account for (a) temperature-dependent behavior for both protonated and deuterated solvents, and (b) temperature-dependent changes in H-bonding in water.

In short, water is a "problematic" solvent, and to properly address the singlet oxygen problem, Richardson and colleagues must go beyond a single temperature study of this H-bonding solvent. A strong case would be made if they also consider benzene or toluene as tunneling candidates, and hexane, for example, as a non-tunneling candidate. I recognize this is a challenge that could take several years. Thus, the present report needs to be modified accordingly.

1. Conclusions from the 2016 PCCP paper (their reference 11):

Pronounced temperature-dependent changes in the singlet oxygen lifetime (non-radiative decay) are reported.

An activation barrier is proposed, and experimental data are presented that imply tunneling may be involved. Specifically, the temperature dependent solvent H/D effect (unique behavior for water, benzene, toluene, and methanol) implies that tunneling through the barrier may play a role in these solvents, but not in other solvents.

Water is a complicated solvent for this test of tunneling.

2. Conclusions from the 2020 JPC paper (their reference 20):

A model, called PaAD, is developed that does NOT involve tunneling and that accounts for the experimental temperature dependent data of most solvents.

As a preface to PaAD, a one-dimensional tunneling model was used to computationally model temperature-dependent behavior, and it did not represent the experimental data well.

The following question was posed: could a multi-dimensional model improve the calculation of a temperature-dependent tunneling phenomenon? This arguably set the stage for the present

submission to Nature Communications.

3. Conclusions from the 2022 JACS paper (their reference 35):

Extended the PaAD model developed in the 2020 paper, moving very close to a complete ab initio solution that does not involve tunneling.

Demonstrated that PaAD worked very well for a large range of solvents, including benzene and toluene (D and H) for which tunneling had earlier been suggested.

The problems associated with water and H-bonding solvents were highlighted, focusing on temperature-dependent changes in the extent of H-bonding.

Reviewer #2 (Remarks to the Author):

This work assesses the nonradiative decay of singlet O₂ to triplet O₂ in the presence of water. It is proposed a tunneling mechanism that speeds up the process by 27 orders of magnitude at room temperature. The tunneling calculations were performed by the instanton method. There is no doubt that the idea of reporting an increase in the process by such a large amount due to quantum effects is very appealing but hard to believe.

I am not aware of any reaction that at room temperature is accelerated by this factor. I am more in favor of Jensen and coworkers' (JPCB, 2020, 2245) interpretation, that is, '... an increase in the H/D solvent isotope effect with a decrease in temperature may not constitute definitive proof for tunneling, and an alternative mechanism may be equally valid. ... the solvent isotope effect is principally determined by the weak coupling term, but its temperature dependence is determined by the Marcus term.' I went carefully through that reference and there is no need to invoke tunneling effects to obtain temperature-dependent KIEs. Richardson and coworkers do not provide instanton calculations at other temperatures, so it is not possible to appreciate the temperature dependence. This check is mandatory to offer some credibility to the proposed mechanism.

Jensen and coworkers employed a simple tunneling model that underestimates the effect, so they concluded that tunneling was unimportant. Richardson and coworkers included multidimensionality to the problem, and I agree with them that corner-cutting effects are important but are they so relevant that they increase the reaction coefficient by 27 orders of magnitude?

There are solvents for which the nonradiative rate constant decreases with increasing temperature, how does a tunneling-based mechanism fit into that situation?

There is an additional aspect that needs careful consideration. It is assumed that the KIE comes from the second step of the reaction, but K_c is not calculated. It should not be hard to estimate this equilibrium constant to make sure its contribution to the KIE is negligible.

Despite the above comments, I find that the manuscript is very original and rigorous. It may offer an alternative explanation for the nonradiative decay of O₂. In my opinion, the work deserves publication in Nature Communications if the authors can offer some additional indicator that the process occurs by tunneling (for instance, correct temperature dependence).

Reviewer #3 (Remarks to the Author):

The authors address the mechanism of decay of singlet ($^1\Delta_g$) oxygen to the triplet ground state in the presence of water as a solvent using theoretical and computational methods.

This is a fundamental process of widespread importance across different branches of chemistry - therefore relevant for a large research community - but still far from being fully understood.

In particular, the authors perform multi-reference electronic structure and decay rate constant calculations for a O₂-H₂O adduct. Classical nonadiabatic transition state theory applied to this model predicts essentially no decay, whereas incorporating quantum tunnelling in the description allows reproducing the order of magnitude of the experimental decay rate, along with an excellent match for the kinetic isotope effect.

The modelling of the singlet-triplet decay via tunnelling is based on a new extension of the instanton theory, which overcomes the limitation of the existing theories in the case of Marcus inverted regime.

Due to the relevance of the application and the potential impact of the new theoretical approaches, this work certainly aligns with the standards for publication in Nat. Comm.

However, I believe there is still margin to improve the manuscript, make it more understandable for a general readership, and discuss some aspects more extensively, before the publication.

Below, I list some points that I believe require further work from the authors:

1 - The result section "Instanton tunnelling mechanism" is rather hard to follow, without fully reading the "Methods section" (where the theoretical details are indeed very nicely explained). I suggest to try to move some key theoretical results in the results section, even if they are not fully derived. For example, if Eq. (12a) is introduced, it becomes then relatively easy to explain the essence of the stationary-action and stationary-flux methods, the fact that the Θ factor can be zero, and that the branching-point method overcomes this problem.

2 - There are different factors, neglected by the ab initio model, that could affect the comparison with the experiment, although it is unlikely that they invalidate the proposed tunnelling mechanism. An important one is the presence of hydrogen-bonding networks that, as discussed in Ref. 20, might make the 1:1 O₂:water model unrealistic. This type of effect is not accounted for by PCM theory.

Indeed since the H atom stretches are predicted to have an important role in the tunnelling mechanism, the impact of further H-bonded molecules should be discussed in the manuscript.

3 - Another ab initio aspect is the dynamical electron correlation which is not accounted for by standard MCSCF calculation. A possible way to check if this affects the prediction is to perform additional electronic structure calculations (e.g. at the MCQDPT level) at the hopping points, and compare the energy gaps and spin-orbit couplings.

4 - The stationary-flux (SF) branching point (BP) methods give very similar results. In a more general case, it is unknown how many branching points one needs to account for, while, on the other hand, the stationary-flux method avoids branching points by construction and seems quite accurate. Why should this approach not be the preferred one?

I see that the discrepancy between the curves of Fig. 7(a) goes in favor of the branching-point approach. However, since $c_{ff}(\tau)$ is anyhow evaluated numerically, would it be possible to derive a better approximation for the stationary-flux correlation function, such as $c_{ff}(\tau_{SF} + i t) = c_{ff}(\tau_{SF}) * \exp(-\mu_{SF} t^2) * (1 + a_3 * t^3 + a_4 * t^4 + ..)$?

The integral for the transition rate would still be analytic. Is this a practical - and perhaps more convenient - route?

Finally some minor aspects:

5 - I suggest to extend Fig. (2) including, for the different structures, the bond distances, electronic energies, spin-orbit couplings, etc.

6 - Which geometry is used to evaluate the spin-orbit coupling Δ for the BP and the SF methods? From the data in the supplementary information it seems that different geometries are used, and indeed the differences in Δ^2 almost fully explain the differences between the SF and BP rate constants (on the other hand the assumption is that this quantity is geometry-

independent...)

**Department of Chemistry and Applied Bio-
sciences**

ETH Zurich
Prof. Dr. Jeremy O. Richardson
Theoretical Molecular Quantum Dynamics
HCI D 267.3
Vladimir-Prelog-Weg 2
8093 Zurich, Switzerland

Editors and Reviewers
Nature Communications

Phone +41 44 633 46 36
jeremy.richardson@phys.chem.ethz.ch

Zurich, 6th March 2024

Manuscript Revision

Dear Editors and Reviewers,

We thank the referees for their comments and provide replies here. We have updated the text of our paper to clarify a number of points for the reader and to provide a more detailed comparison of our new theory with previous models. We hope that you will now consider our paper ready for publication.

Response to Reviewer 1

This paper addresses an important topic, and the authors present their theoretical and computational work in the context of a long-standing important and challenging problem: the solvent dependent nonradiative deactivation of singlet oxygen.

However, in their presentation, I believe the authors put "the cart before the horse" by providing an extensive preface on singlet oxygen, leading the reader to infer that their work resolves the singlet oxygen deactivation problem. This sentiment is clearly expressed in the abstract where the authors indicate that their approach "holds the key to elucidating the reaction mechanism."

The authors indicate that the impetus for their study is embodied in two papers (their references 11 and 20) in which the possible contribution of tunneling in the deactivation of singlet oxygen was both experimentally and computationally considered. Outlined below are detailed comments on these published papers, as they apply to the present submission to Nature Communications. Of note is the fact that (a) evidence against a tunneling contribution was eventually provided, and (b) a satisfactory computational model, so-called PaAD, that does not consider tunneling, accounts for all the temperature dependent experimental data.

Despite the rather extensive work from Richardson's group reported in this manuscript, they do not satisfactorily "solve" the singlet oxygen problem, nor do they provide substantive evidence that argues against the results of the PaAD model described in their references 20 and 35.

We thank the reviewer for their comments on putting our work into the context of what is already understood about singlet oxygen deactivation. From the start, we wish to make it clear that it is not our intention to argue *against* the results of the PaAD model, but rather seek to ground it in a more rigorous foundation.

It is important to point out that the PaAD model (as defined in Thorning et al. JACS 2022) contains two or three parameters which are fit to experiment (separately for H₂O and D₂O), whereas our ab initio instanton theory approach is completely parameter free. There is no doubt that both approaches have their role to play in the

scientific endeavour to understand and predict the oxygen deactivation lifetimes. The advantage of the PaAD model is in its simplicity, that it highlights only the most important details and can be evaluated with a handful of easily accessible computational or experimental observables. On the other hand, the more rigorous instanton theory can be used in cases where one has little intuition to obtain understanding without biasing the result with preconceptions. In this particular case, we can use instanton theory to explain the success of the PaAD model, to provide a more rigorous justification, to search for its limitations and give suggestions for how it can be further improved.

The PaAD model is a weighted average of Jortner's weak-coupling theory of nonradiative processes and a modified version of Marcus theory. Both of these methods are based on Fermi's golden rule. The former takes a steepest-descent approximation to evaluate the Franck–Condon overlaps within a low-temperature harmonic approximation. The latter uses a classical approximation to describe a thermally activated process. It was the addition of the latter term which led to the great success of PaAD in being able to describe the temperature dependence in certain solvents.

Instanton theory is also based on Fermi's golden rule, but takes fewer approximations than either Jortner's or Marcus' theories. In fact, it encompasses both theories as limiting cases. The great advantage of our instanton approach is therefore that it automatically determines whether the system is in Jortner's or Marcus' regime, or something in between. In future work, we intend to apply instanton theory to singlet oxygen in other solvents and expect to find that it recovers much of the behaviour of the PaAD model, but without requiring parameters to be determined from experiment. This will lead us closer to a full understanding of the process.

One might think there is a contradiction between instanton theory, which predicts tunnelling effects, and the PaAD model, which reports that they are not necessary. This is not the case. It is important to note that Jortner's theory (which is typically the dominant contribution to the PaAD model) includes low-temperature Frank–Condon factors. Within instanton theory, which is based on path integrals rather than wavefunctions, Franck–Condon factors are captured using tunnelling pathways, i.e. paths with total energy below that of the potential energy. Whether we decide call this tunnelling or simply Franck–Condon overlaps is a matter of convention, but what cannot be denied is that a classically-forbidden process occurs with a strong isotope effect, and that the classically-allowed process is orders of magnitude slower than the quantum-mechanical reaction.

Here, it is worth mentioning how much more insight can be obtained from the instanton method than from Jortner's theory. The instanton pathway gives a clear picture of which atoms are involved with direct implications for kinetic isotope effects. In Jortner's theory, an effective harmonic oscillator is used to model the ground and excited states (with the same frequencies in both). There is thus little mechanistic insight and it cannot be applied to a realistic anharmonic molecule without making nonunique choices for the parameterization of the effective harmonic oscillator (e.g. should one use the normal modes of the singlet state or triplet state or an average of the two?).

In fact, when the PaAD model was first proposed, it was argued that the curvature of the excited-state PES would be significantly different from that of the ground state leading to a break-down of Jortner's theory. This is exactly what we find in our instanton calculations. However, whereas the PaAD model dealt with this problem by introducing a second term based on Marcus theory, the instanton approach retains a unified description which treats the different curvatures explicitly.

Importantly, as the PaAD parameters are fit separately for each solvent, it cannot be used to predict the lifetime in a new solvent, nor to predict kinetic isotope effects (KIEs). In contrast, we have demonstrated that instanton theory can give a very accurate prediction of kinetic isotope effects of singlet oxygen in water.

In the revised manuscript, we provide a discussion of the connection between instanton theory and the PaAD model.

Rather, Richardson and colleagues should re-cast their presentation to say that they have developed an approach to deal with tunneling through barriers in the Marcus inverted region, which can be relevant for a host of reactions, including the solvent-mediated nonradiative deactivation of singlet oxygen. They should then mention that, as noted in references 11, 20 and 35, to properly address the singlet oxygen problem, their computations must eventually account for (a) temperature-dependent behavior for both protonated and deuterated solvents, and (b) temperature-dependent changes in H-bonding in water.

In short, water is a "problematic" solvent, and to properly address the singlet oxygen problem, Richardson and colleagues must go beyond a single temperature study of this H-bonding solvent. A strong case would be made if they also consider benzene or toluene as tunneling candidates, and hexane, for example, as a non-tunneling candidate. I recognize this is a challenge that could take several years. Thus, the present report needs to be modified accordingly.

The reviewer is of course quite right that the branch-point instanton method is a powerful approach with wide ranging implications. It will be relevant for much more than just singlet oxygen and we intend to explore other nonradiative processes in future work.

We also agree that there is further work to be done in applying our instanton theory to singlet oxygen in other solvents to fully "solve" the problem of singlet oxygen. However, we stand by our statement that the method "holds the key to elucidating the reaction mechanism," and we believe that this approach is the right direction to obtain the "ultimate goal: an accurate and predictive *ab initio* model to account for the effect of solvent on the lifetime of singlet oxygen" (Thorning et al. JACS 2022).

It seems to us that one of the main complications of studying water with the PaAD model (as expressed in Thorning et al. JACS 2022) is that it is not obvious which OH frequency to use within the model due to the H-bonding leading to a broad IR spectrum. This would not cause a complication for an instanton calculation of an O₂ molecule surrounded by a single water molecule or a cluster of water molecules. In instanton theory, one does not need to isolate an individual frequency, but simply works with the full set of frequencies obtained from the full-dimensional simulation. The method would automatically take care of the fact that some modes are more strongly coupled to the O₂ than others. There is thus hope that such a simulation would provide a reliable prediction and a clearer understanding of the effects of H-bonding. The fact that Thorning et al. (JACS 2022) find evidence that the rate-limiting step depends only on local interactions gives hope that reasonably small clusters will give converged results which can be assumed to be good approximations for the reaction in solution.

We have now included temperature-dependent data for singlet oxygen relaxation in water. It seems that the reaction rate constant, k_2 , is almost temperature independent for O₂ in H₂O, as is expected for a deep-tunnelling process. Therefore, the dominant effect to the temperature dependence must come from the equilibrium constant K_c . Our crude approximation to this does indeed predict that the rate increases with temperature, in reasonable agreement with experiment. A more rigorous simulation of the equilibrium constant would be possible in principle using a path-integral molecular dynamics simulation of singlet O₂ in liquid water as long as useful collective variable can be identified for the complexation. This calculation of K_c would be combined with the instanton calculation of k_2 to give an *ab initio* prediction of the overall lifetime. This is, however, beyond the scope of the present work, which is instead focused on the mechanism of the rate-determining step.

Without access to reliable calculations of K_c , one test we can carry out is to fit the function (using two parameters corresponding to the entropy and enthalpy changes of complexation). In our revised supplementary information, we show that this two-parameter fit is able to recover the experimental data, demonstrating that our results are at least not inconsistent with the measured data. A better test (which involves no fitting parameters at all) is to evaluate the kinetic isotope effect (KIE), as the contribution of the unknown K_c is expected to cancel here. In the revised main text, we show that the temperature-dependence of the KIE is in excellent agreement with the experimental measurements. This provides more evidence that our predicted mechanisms are physical

and our predictions reliable.

1. Conclusions from the 2016 PCCP paper (their reference 11):

Pronounced temperature-dependent changes in the singlet oxygen lifetime (non-radiative decay) are reported.

An activation barrier is proposed, and experimental data are presented that imply tunneling may be involved. Specifically, the temperature dependent solvent H/D effect (unique behavior for water, benzene, toluene, and methanol) implies that tunneling through the barrier may play a role in these solvents, but not in other solvents.

Water is a complicated solvent for this test of tunneling.

2. Conclusions from the 2020 JPC paper (their reference 20):

A model, called PaAD, is developed that does NOT involve tunneling and that accounts for the experimental temperature dependent data of most solvents.

As a preface to PaAD, a one-dimensional tunneling model was used to computationally model temperature-dependent behavior, and it did not represent the experimental data well.

The following question was posed: could a multi-dimensional model improve the calculation of a temperature-dependent tunneling phenomenon? This arguably set the stage for the present submission to Nature Communications.

3. Conclusions from the 2022 JACS paper (their reference 35):

Extended the PaAD model developed in the 2020 paper, moving very close to a complete ab initio solution that does not involve tunneling.

Demonstrated that PaAD worked very well for a large range of solvents, including benzene and toluene (D and H) for which tunneling had earlier been suggested.

The problems associated with water and H-bonding solvents were highlighted, focusing on temperature-dependent changes in the extent of H-bonding.

We are in agreement with the majority of these conclusions from previous work, in particular:

1. Instanton theory also predicts temperature-dependent changes in rates and kinetic isotope effects, as we now show in the revised manuscript. It reduces to Marcus theory in certain limits and will thus also be able to describe the activation barrier and tunnelling effects, where appropriate.
2. We agree that there is no tunnelling through the MECP, but do find tunnelling elsewhere, which is also implicitly included within the PaAD model through the Frank–Condon factors. Indeed it seems that it is important to treat the multidimensional nature of the system.
3. In all these cases where the PaAD model worked well, we expect that instanton theory can be used to provide a more rigorous justification for the model and more insight into the mechanism. It provides extra information which is not available to the PaAD model and which is needed to push towards our ultimate goal of obtaining the complete ab initio solution to the singlet oxygen problem.

To reiterate, the key point of our work is not that we overturn the previous models of oxygen deactivation, but rather that we provide a fully ab initio computational method to make predictions and provide novel mechanistic

insight.

Response to Reviewer 2

This work assesses the nonradiative decay of singlet O₂ to triplet O₂ in the presence of water. It is proposed a tunneling mechanism that speeds up the process by 27 orders of magnitude at room temperature. The tunneling calculations were performed by the instanton method. There is no doubt that the idea of reporting an increase in the process by such a large amount due to quantum effects is very appealing but hard to believe.

I am not aware of any reaction that at room temperature is accelerated by this factor. I am more in favor of Jensen and coworkers' (JPCB, 2020, 2245) interpretation, that is, '... an increase in the H/D solvent isotope effect with a decrease in temperature may not constitute definitive proof for tunneling, and an alternative mechanism may be equally valid. ... the solvent isotope effect is principally determined by the weak coupling term, but its temperature dependence is determined by the Marcus term.' I went carefully through that reference and there is no need to invoke tunneling effects to obtain temperature-dependent KIEs. Richardson and coworkers do not provide instanton calculations at other temperatures, so it is not possible to appreciate the temperature dependence. This check is mandatory to offer some credibility to the proposed mechanism.

Jensen and coworkers employed a simple tunneling model that underestimates the effect, so they concluded that tunneling was unimportant. Richardson and coworkers included multidimensionality to the problem, and I agree with them that corner-cutting effects are important but are they so relevant that they increase the reaction coefficient by 27 orders of magnitude?

Yes, it is possible to find such enormous tunnelling effects in room-temperature chemical reactions! The fact that the reviewer finds this so unbelievable is what makes our study so interesting and shows how it changes the way we should think about tunnelling.

To understand how this is possible, one needs to realize that our mechanism is fundamentally different from the usual one-dimensional WKB approaches used by Jensen and coworkers. The concept of "corner cutting" is perhaps not powerful enough to explain these results. Instead let us consider as an analogy an adiabatic chemical reaction with two completely different mechanisms. The first mechanism has very high and wide barrier and the second mechanism has an even higher barrier but one which is very narrow. In a classical theory, the rate would be dominated by the first mechanism, but it would be extraordinarily slow. In a quantum theory, the rate would be dominated by the second mechanism, which may be quite fast. By tuning the parameters of the model, we could make the ratio between the two rates arbitrarily large.

The results of our paper are analogous to this simple model. The first mechanism passes through the MECP, which is very high in energy and for which Jensen and coworkers have already shown that tunnelling effects are negligible. The second mechanism is described by the branch-point instanton. This is not an energy-conserving processes and thus formally has no classical analogue, but within quantum mechanics (which is here well approximated by instanton theory) the rate is relatively fast. This explains why we are able to predict such large ratios between the rates calculated by instanton theory and within a classical picture.

As discussed in our reply to the first reviewer, we now present results at a range of temperatures. Our predictions are in excellent agreement with the temperature dependence of the experimental kinetic isotope effects.

There are solvents for which the nonradiative rate constant decreases with increasing temperature, how does a tunneling-based mechanism fit into that situation?

To answer this question, it is important to appreciate the differences between the various rate constants discussed in this work and in the literature. With instanton theory, we directly compute k_2 , the rate constant for a

reaction to occur within the 1:1 complex. However, in Bregnhøj et al. (PCCP 2016), the rate constant plotted is $k_{nr} = K_c k_2$, where K_c is the equilibrium constant for the formation of the complex. The reviewer is correct that ordinarily, tunnelling rates (k_2) show a monotonic increase with temperature. However, at low temperatures, they may become temperature independent. In fact, in $O_2 \cdots H_2O$, we find that the rate constant k_2 is almost independent of temperature (a clear indication of deep tunnelling). In Bregnhøj et al. (PCCP 2016), k_{nr} was found to decrease with increasing temperature in certain solvents, such as acetone. This is not inconsistent with our findings and we can provide two possible explanations. Future work will be necessary to determine which is correct (or something in between). If strong tunnelling effects occur in acetone, similarly to those we have found in water, then k_2 will be constant, and the temperature dependence of k_{nr} will be determined by K_c . Considering van 't Hoff's equation $d \ln K_c / d(1/T) = -\Delta_r H / R$, the inverse effect would thus appear in the calculation of K_c whenever the enthalpy of the 1:1 complex is lower than that of the non-complexed solution, i.e. whenever the complexation process is exothermic. Presumably, in order to form the complex in water, the hydrogen-bonding network must be disrupted such that the process is endothermic, whereas in acetone, there is no such penalty making it exothermic. Alternatively if tunnelling effects are negligible, the rate will be determined by an Arrhenius relationship (such as Marcus theory), for which an inverse Arrhenius behaviour is possible. In fact, this aspect could already be accounted for theoretically with the simple model from Thorning et al. (JPCB 2020), as they found the vibrationally resonant point to lie below the energy of the separated molecules, making the effective activation barrier negative.

There is an additional aspect that needs careful consideration. It is assumed that the KIE comes from the second step of the reaction, but K_c is not calculated. It should not be hard to estimate this equilibrium constant to make sure its contribution to the KIE is negligible.

In the high-temperature limit, where nuclear quantum effects can be neglected, the equilibrium constant will be identical for H_2O and D_2O , because classical statistical mechanics is rigorously independent of isotope effects. We effectively assume that this limit continues to hold approximately at room temperature. Note that this does not contradict our finding of strong tunnelling effects in the rate constant k_2 . This is because K_c is dominated by the weak intermolecular modes (with frequencies $\hbar\omega \ll k_B T$), whereas the tunnelling pathway which dominates k_2 excites strong intramolecular modes (with $\hbar\omega \gg k_B T$). In the supplementary information, we have now included estimates of K_c based on our crude gas-phase model (employing quantum harmonic oscillators), which clearly demonstrate the weak dependence on isotope.

A more rigorous calculation of K_c would require a converged path-integral molecular dynamics simulation of singlet O_2 in liquid water, which is beyond the scope of this study.

Despite the above comments, I find that the manuscript is very original and rigorous. It may offer an alternative explanation for the nonradiative decay of O_2 . In my opinion, the work deserves publication in Nature Communications if the authors can offer some additional indicator that the process occurs by tunneling (for instance, correct temperature dependence).

We appreciate the reviewer's positive remarks on our work and we hope that you will agree that our additional calculations at various temperatures lend further weight to the validity of the tunnelling mechanisms proposed.

Response to Reviewer 3

The authors address the mechanism of decay of singlet ($1\Delta_g$) oxygen to the triplet ground state in the presence of water as a solvent using theoretical and computational methods.

This is a fundamental process of widespread importance across different branches of chemistry - therefore relevant for a large research community - but still far from being fully understood.

In particular, the authors perform multi-reference electronic structure and decay rate constant calculations for a O_2 - H_2O adduct. Classical nonadiabatic transition state theory applied to this

Manuscript Revision

model predicts essentially no decay, whereas incorporating quantum tunnelling in the description allows reproducing the order of magnitude of the experimental decay rate, along with an excellent match for the kinetic isotope effect.

The modelling of the singlet-triplet decay via tunnelling is based on a new extension of the instanton theory, which overcomes the limitation of the existing theories in the case of Marcus inverted regime.

Due to the relevance of the application and the potential impact of the new theoretical approaches, this work certainly aligns with the standards for publication in Nat. Comm.

However, I believe there is still margin to improve the manuscript, make it more understandable for a general readership, and discuss some aspects more extensively, before the publication.

Below, I list some points that I believe require further work from the authors:

1 - The result section "Instanton tunnelling mechanism" is rather hard to follow, without fully reading the "Methods section" (where the theoretical details are indeed very nicely explained). I suggest to try to move some key theoretical results in the results section, even if they are not fully derived. For example, if Eq. (12a) is introduced, it becomes then relatively easy to explain the essence of the stationary-action and stationary-flux methods, the fact that the Θ factor can be zero, and that the branching-point method overcomes this problem.

We thank the reviewer for their good suggestion. We have added an extra paragraph to give the key equation from the Methods section to help readers follow the main arguments without fully reading the "Methods" section.

2 - There are different factors, neglected by the ab initio model, that could affect the comparison with the experiment, although it is unlikely that they invalidate the proposed tunnelling mechanism. An important one is the presence of hydrogen-bonding networks that, as discussed in Ref. 20, might make the 1:1 O₂:water model unrealistic. This type of effect is not accounted for by PCM theory.

Indeed since the H atom stretches are predicted to have an important role in the tunnelling mechanism, the impact of further H-bonded molecules should be discussed in the manuscript.

We agree with the reviewer that the consideration of only an isolated O₂...H₂O complex is a limitation of our current approach, but it is a limitation which is shared by the previous approaches of Thorning et al. (JCPB 2020). Note, however, that our first-principles instanton theory is also applicable to study a cluster of O₂ with multiple H₂O molecules, and so in future work, one could systematically test the validity of the approximation and determine the dependence of many-body effects on the lifetime. We have added a short discussion of this point with suggestions for future work.

3 - Another ab initio aspect is the dynamical electron correlation which is not accounted for by standard MCSCF calculation. A possible way to check if this affects the prediction is to perform additional electronic structure calculations (e.g. at the MCQDPT level) at the hopping points, and compare the energy gaps and spin-orbit couplings.

Following the suggestion of the reviewer, we now present energies from single-point MRMP2 calculations at relevant geometries in the SI. Except for an unimportant overall shift, it seems that the MCSCF captures the shapes reasonably well, although there are of course some differences. It is, however, hard to estimate the effect that the differences will have on the rate without rerunning the entire calculations using MRMP2. We rely on the usual assumption that most of the error will cancel out in the calculation of the kinetic isotope effect.

As reported in the Methods section, the spin-orbit coupling changes by only 4% using SO-MCQDPT.

4 - The stationary-flux (SF) branching point (BP) methods give very similar results. In a more general case, it is unknown how many branching points one needs to account for, while, on the other hand, the stationary-flux method avoids branching points by construction and seems quite accurate. Why should this approach not be the preferred one?

I see that the discrepancy between the curves of Fig. 7(a) goes in favor of the branching-point approach. However, since $c_{ff}(\tau)$ is anyhow evaluated numerically, would it be possible to derive a better approximation for the stationary-flux correlation function, such as $c_{ff}(\tau_{SF} + i t) = c_{ff}(\tau_{SF}) * \exp(-\mu_{SF} t^2) * (1 + a_3 * t^3 + a_4 * t^4 + \dots)$?

The integral for the transition rate would still be analytic. Is this a practical - and perhaps more convenient - route?

We have a number of reasons for preferring the BP approach over SF.

- (a) It is more mathematically rigorous as it is formally a semiclassical approximation which becomes exact in the $\hbar \rightarrow 0$ limit (like the original steepest-descent instanton theory).
- (b) As a consequence of the more rigorous derivation, the accuracy of BP is clearly superior in the model system (Fig 8).
- (c) To calculate the BP rate, one simply needs to extract the first derivative of ϕ and Θ at τ_{BP} , whereas for the SF method, one needs to numerically determine the second derivative of a function, which is more susceptible to numerical errors and more difficult to define a unique result independent of the exact details of the fitting algorithms.
- (d) Although we have not addressed the point in this paper (as it was not necessary for the systems studied), extending the BP theory to include multiple branch points appears to be quite possible, but is left for future work.
- (e) We would actually expect the SF method to become less accurate in cases with multiple branch points. The reviewer's suggestion of including higher-order derivatives is in principle a good idea to fix this, but may not be very numerically stable as we only have access to derivatives with respect to τ and would have to analytically continue them to get high-order derivatives with respect to t , which may exacerbate small fitting errors. Nonetheless, we will test this idea in future work.

Finally some minor aspects:

5 - I suggest to extend Fig. (2) including, for the different structures, the bond distances, electronic energies, spin-orbit couplings, etc.

We chose not to follow this suggestion so as to avoid cluttering the figure. Key geometrical information is already available in Fig. 4 and energies are given in the supplementary material.

6 - Which geometry is used to evaluate the spin-orbit coupling Δ for the BP and and the SF methods? From the data in the supplementary information it seems that different geometries are used, and indeed the differences in Δ^2 almost fully explain the differences between the the SF and BP rate constants (on the other hand the assumption is that this quantity is geometry-independent...)

In all our instanton theories, the spin-orbit coupling is measured at the hopping point. As the hopping point appears at slightly different configurations in the BP and SF methods, the value of Δ used is slightly different in the two cases. Note that in our derivation we have *not* assumed that Δ is geometry-independent, but rather that it is slowly varying on a length scale determined by the fluctuations around the instanton path. Higher-order terms could in principle be included to account for the rate of change of Δ , but these would be associated with an extra factor of \hbar . Instanton theory is an asymptotic approximation which becomes exact in the $\hbar \rightarrow 0$ limit,

Manuscript Revision

and thus these extra terms are subdominant.

The reviewer makes an interesting observation that Δ^2 accounts for a large part of the difference between the SF and BP methods. However, it is by no means the only contributing factor. For instance, for $\text{O}_2 \cdots \text{D}_2\text{O}$ at 300 K (cc-pVDZ level with $N = 512$), the ratio between Δ^2 is 1.105, whereas the ratio of the rate constants is 1.165. In other words, other effects contribute a factor of $1.165/1.105=1.054$. If the values of Δ were wildly different in the two cases, it would imply a breakdown of our assumptions. However, the 10% variation observed here is not a problem, especially when considering the other approximations made.

Yours sincerely

Jeremy Richardson

Articles referred to in this response letter:

- M. Bregnhøj, M. Westberg, F. Jensen, and P. R. Ogilby, "Solvent-dependent singlet oxygen lifetimes: temperature effects implicate tunneling and charge-transfer interactions, *Phys. Chem. Chem. Phys.*, vol. 18, pp. 22946–22961, (2016).
- F. Thorning, F. Jensen, and P. R. Ogilby, "Modeling the effect of solvents on nonradiative singlet oxygen deactivation: Going beyond weak coupling in intermolecular electronic-to-vibrational energy transfer," *J. Phys. Chem. B*, vol. 124, pp. 2245–2254, (2020).
- F. Thorning, P. Henke, and P. R. Ogilby, "Perturbed and activated decay: The lifetime of singlet oxygen in liquid organic solvents," *J. Am. Chem. Soc.*, vol. 144, pp. 10902–10911, (2022).

Reviewer #2 (Remarks to the Author):

The authors have thoroughly addressed all the comments raised by the reviewers. While some may argue that invoking tunneling to explain this reaction mechanism is unnecessary, the innovative theoretical interpretation by Richardson and colleagues should not be dismissed. After carefully weighing the advantages and disadvantages of the method, I believe that this manuscript should be published in Nature Communications.

Reviewer #3 (Remarks to the Author):

In the revised manuscript Richardson and coworkers address rather well the concerns and queries raised by the reviewers, resulting in a significantly enhanced and clearer manuscript. They also thoroughly discuss the limitations of the present study, in particular the exclusive focus on the 1:1 solute solvent complex, and the realistic experimental comparisons only possible for the kinetic isotope effect (KIE).

Particularly noteworthy in the new version is the study of the temperature-dependence of the KIE. Despite the limitations of the 1:1 model, that does not properly account for the H-bonding network, the experimental trend is convincingly captured.

Furthermore, the present theoretical formulation - entirely from first-principles - is now discussed in connection with the previously proposed PaAD theory, that is nearly but not completely *ab initio*. I agree with the authors that tunnelling effects are implicitly present in the PaAD model via Franck-Condon factors. Therefore, the present instanton model does not invalidate, but rather improves previous theories, giving mechanistic insights that were previously not fully accessible, and pointing out that a multidimensional view of the mechanism is necessary.

Of course, this work does not mark the end of the efforts to predict O₂ deactivation rates from first principles. However, the study presents an entirely *ab initio* approach to model the rate-limiting step of the O₂ decay in water, that is in principle applicable across various solvents. It is challenging for me to envision another parameter-free methodology that has the same computational price and simultaneously accounts for multi-dimensional quantum effects with fewer approximations.

Considering also that the developed computational method is generally applicable to other radiationless phenomena, I recommend the publication in Nature Communications.

Reviewer #4 (Remarks to the Author):

I have not reviewed the original manuscript.

Reading the revised manuscript, I get the impression that this is a significant new development of using instanton theory to model tunneling processes in the inverted Marcus region. The application to the 1O₂ - H₂O system can be considered as an illustration for one of the smallest systems where this is of interest. Although this may be a system where the connection to experimental results is the least transparent, the agreement with experimental results both in terms of absolute rates and kinetic isotope effects is impressive. If the tunneling enhancement of 27 order of magnitude stand up to further refinements of the system model, this is indeed remarkable. I consider the authors responses to the previous reviewer comments as being of high quality.

I recommend publication in its present form.